# Spastin recovery in hereditary spastic paraplegia by preventing neddylation-dependent degradation

Francesca Sardina[1] , Alessandra Pisciottani[1], Manuela Ferrara[1], Davide Valente[1,5] , Marialuisa Casella[2], Marco Crescenzi[2], Angelo Peschiaroli[3], Carlo Casali[4], Silvia Soddu[5], Andrew J Grierson[6], Cinzia Rinaldo[1,5]

Hereditary Spastic Paraplegia (HSP) is a neurodegenerative disease most commonly caused by autosomal dominant mutations in the *SPG4* gene encoding the microtubule-severing protein spastin. We hypothesise that *SPG4*-HSP is attributable to reduced spastin function because of haploinsufficiency; thus, therapeutic approaches which elevate levels of the wild-type spastin allele may be an effective therapy. However, until now, how spastin levels are regulated is largely unknown. Here, we show that the kinase HIPK2 regulates spastin protein levels in proliferating cells, in differentiated neurons and in vivo. Our work reveals that HIPK2-mediated phosphorylation of spastin at S268 inhibits spastin K48-polyubiquitination at K554 and prevents its neddylation-dependent proteasomal degradation. In a spastin RNAi neuronal cell model, overexpression of HIPK2, or inhibition of neddylation, restores spastin levels and rescues neurite defects. Notably, we demonstrate that spastin levels can be restored pharmacologically by inhibiting its neddylation-mediated degradation in neurons derived from a spastin mouse model of HSP and in patient-derived cells, thus revealing novel therapeutic targets for the treatment of *SPG4*-HSP.

## Introduction

Spastin is an AAA ATPase microtubule (MT)-severing enzyme that regulates cytoskeleton rearrangement associated with membrane remodeling. It is a critical regulator in cytokinesis and nuclear envelope resealing during cell division and it is also involved in intracellular traffic (1, 2, 3). In neurons, spastin has key roles in axonal transport and regeneration (4, 5, 6). This enzyme functions as hexamers that drive the remodeling and severing of MT by tugging the C-terminal tail of tubulin through the central pore of the hexamer (7, 8, 9). Thus far, four spastin isoforms have been identified, M1 (68 kD), a shorter isoform lacking the first 86 aa (M87,

60 kD), and splice variants of both of these, excluding exon 4 (M1Δ4, 64 kD, and M87Δ4, 55 kD). M1 and M87 are synthesized simultaneously but at largely different levels because the first AUG is surrounded by a weak Kozak sequence resulting in a preferred translation from the second AUG (10, 11). Splice variants are the least abundant isoforms, for which no specific functions have been reported thus far.

It has been suggested that precise regulation of spastin is critical to prevent its dysfunction (5, 6). *SPG4* mutations are the most common cause of autosomal dominant Hereditary Spastic Paraplegia (HSP), a neurodegenerative disease characterized by progressive spasticity of the lower limbs (12, 13, 14, 15, 16). Because most of the *SPG4* pathogenic mutations are nonsense, frameshift or large insertion/deletion mutations that are generally associated with nonsense mediated degradation of mRNA, haploinsufficiency is considered one of the main molecular mechanisms of the disease (17, 18, 19). In a subset of patients carrying missense mutations dominant-negative effect, threshold-effect-model and gain-of-function appear to be relevant (16, 20, 21). Axonal swellings, characterized by aberrant accumulation of neurofilaments and mitochondria, and abnormal organelle distribution/trafficking are the hallmarks of axonal defects in mouse and human *SPG4*-HSP models (22, 23, 24, 25). Loss-of-function mouse *SPG4*-HSP models show motor behavior deficits and axon trafficking impairment; heterozygous mice show very mild phenotypes compared with the homozygous ones, further supporting a dosage effect (22, 23). Studies on neurons generated from *SPG4*-HSP patient-derived induced Pluripotent Stem cells, reported spastin reduction associated with alterations in neurite morphology, swellings and transport deficits (24, 25). Currently, there is no cure for HSP. Importantly, in patients' neurons a spastin gene dosage-dependent rescue of the defects has been recently reported by lentiviral spastin expression, providing proof of principle that spastin elevating approaches could be a therapeutic strategy (25). However, at present little is known about the molecular mechanisms regulating endogenous spastin protein levels.

[1]Institute of Molecular Biology and Pathology (IBPM), National Research Council (CNR), c/o Sapienza University, Rome, Italy   [2]Core Facilities, Italian National Institute of Health, Rome, Italy   [3]Institute of Translational Pharmacology, CNR, Rome, Italy   [4]Department of Medico-Surgical Sciences and Biotechnologies, University of Rome Sapienza, Latina, Italy   [5]Unit of Cellular Networks and Molecular Therapeutic Targets, IRCCS–Regina Elena National Cancer Institute, Rome, Italy   [6]Sheffield Institute for Translational Neuroscience, University of Sheffield, Sheffield, UK

Correspondence: cinzia.rinaldo@uniroma1.it; francesca.sardina3@gmail.com
Alessandra Pisciottani's present address is Università Vita-Salute San Raffaele, Milano, Italy

Recently, we identified spastin as novel target of the multifunctional kinase HIPK2 (26), that is highly expressed in the nervous system (27, 28). We demonstrated that HIPK2-mediated phosphorylation of spastin at serine 268 is required for its midbody localisation and successful abscission. Here, we show that spastin levels are regulated by HIPK2-mediated phosphorylation and neddylation-dependent degradation, providing evidence that it is possible to restore spastin levels and reduce axonal swelling pathology by targeting these pathways in *SPG4*-HSP models.

# Results

## Spastin protein levels are regulated by HIPK2 at post-transcriptional level

During previous studies we noticed a reduction of spastin protein levels upon HIPK2 depletion in HeLa cells, suggesting that HIPK2 might control spastin protein levels (26). To verify this hypothesis, we performed HIPK2-depletion by RNAi, CRE-Lox or CRISPR/Cas9 recombination in cells of different origin and analysed spastin protein levels with different anti-spastin Abs (Figs 1A and B and S1A). Compared with control cells, reduction of spastin levels was observed in all the HIPK2-deficient cells obtained (Figs 1A and B and S1A) and was rescued after HIPK2 re-expression in HeLa HIPK2-Cas9 cells (Fig 1C). Notably, spastin reduction was observed upon HIPK2 depletion in murine neuron-like NSC34 differentiated cells (Figs 1D and S1B and C), in human SH-SY-5Y neuron-like differentiated cells (data not shown) and in primary cortical neurons explanted from CRE-inducible *Hipk2* KO mice upon infection with an adenovirus expressing the recombinase CRE (Figs 1E and S1D), showing that spastin levels are regulated by HIPK2 in postmitotic differentiated primary neurons.

Consistent with these observations, analysis of spastin expression in cerebral cortex (CC) and striatum (STR) explanted from *Hipk2^{KOF/KOF}* mice, showing strongly reduced HIPK2 levels compared with wild-type (WT) siblings (Fig S1E and F), revealed a significant reduction of the protein levels of all detectable spastin isoforms, whereas spastin mRNA levels were unchanged (Fig 1F and G). This suggests that HIPK2 might also control spastin protein levels in the central nervous system (CNS) in vivo.

As reciprocal approach, we showed that HIPK2 up-regulation, obtained by either overexpression of the exogenous protein or heat shock–mediated stabilization (29) of the endogenous one, was associated with increased spastin expression (Figs 1H and S1G).

Altogether these findings demonstrate that spastin is regulated by HIPK2 in proliferating cells, in differentiated neurons and in vivo.

To investigate HIPK2–mediated regulation of spastin, we analysed both spastin mRNA and protein levels in HIPK2-depleted (siHIPK2) and control cells (siCtr). The decrease in spastin protein expression occurs without significant change at the mRNA level upon HIPK2 depletion (Figs 1I and J and S1H), indicating post-transcriptional regulation of protein levels.

## Spastin is polyubiquitinated at K554R and degraded via proteasome in HIPK2-defective cells

Next, we performed protein degradation tests and observed that spastin protein levels could be rescued by treatment with the proteasome inhibitor MG132 in HIPK2-depleted cells (Fig 2A), suggesting that degradation of spastin is mediated by the ubiquitin proteasome pathway (UPP). Thus, we investigated whether spastin is polyubiquitinated in HIPK2-defective cells. By using an anti-Ub Ab recognising mono- and poly-Ub conjugates, a pattern of slowly migrating spastin-Ub conjugates was detectable after spastin IP in HIPK2-depleted cells post MG132 treatment (Figs 2B and C and S2A). To further validate spastin polyubiquitination, we performed ubiquitination assays in HeLa HIPK2-Cas9 and in Ctr-Cas9 cells, using HA-tagged Ub-WT (Ub-HA) or lysine-free Ub mutant (koUb-HA), which allows only monoubiquitination. IP with normal mouse IgG immunoglobulins was used as negative control (Fig S2B). No ubiquitination signals were observed in cells expressing koUb-HA mutant, whereas a smear of slowly migrating spastin-Ub-HA conjugates accumulated in Ub-WT-expressing cells after MG132 treatment in HIPK2-Cas9 cells (Fig 2D), indicating spastin polyubiquitination. Similar results were obtained also in parental Ctr-Cas9 cells (Fig 2D). Next, we performed a ubiquitination assay using a K48-only Ub mutants (K48-Ub-HA), which allows only K48-linked polyubiquitinated chains, the canonical proteasomal degradation signal. Spastin K48-polyubiquitination was detectable after MG132 treatment in HeLa HIPK2-Cas9 and in Ctr-Cas9 cells (Fig S2C and D). Altogether, these findings indicate the involvement of the UPP pathway in the regulation of spastin, even if we cannot rule out the involvement of other non-proteolytic processes.

Among all the lysines of spastin that are possible sites for Ub conjugation, K554 was reported as a putative site in a large-scale mass spectrometry (MS) screen (https://www.phosphosite.org; (30)). Thus, we focused our attention on this lysine residue which is evolutionary conserved (Fig 2E) and located in the ATPase domain common to all spastin isoforms. We generated a spastin K554 mutant (spastin-K554R), and tested it in ubiquitination assays. Flag-myc tagged spastin-K554R and spastin-WT were expressed in combination with Ub-HA in HIPK2-Cas9 cells. As shown in Figs 2F and S2E, spastin-WT displayed a smear of slowly migrating bands comparable to that of the endogenous protein, whereas spastin-K554R showed a very strong reduction of this pattern, indicating that K554 is a key spastin polyubiquitination site (Fig 2F). Accordingly, when transfected at comparable efficiency, spastin-WT levels were reduced in HIPK2-Cas9 cells compared with HIPK2 proficient cells, whereas spastin-K554R was expressed at similar levels in both cell types, suggesting that spastin-K554R is more resistant to protein degradation than spastin-WT (Fig 2G). These results show that K554 polyubiquitination is required for spastin degradation in HIPK2-defective cells.

## HIPK2 regulates spastin protein levels through its kinase activity

We have demonstrated that HIPK2 binds and phosphorylates spastin at S286 (26), thus we analysed whether HIPK2 regulates spastin protein levels through its kinase activity. Compared with HIPK2-WT–overexpressing cells, no increase in spastin protein levels was observed upon transfection of the kinase-defective HIPK2-K228A mutant (Fig 3A), consistent with a role for HIPK2 kinase activity in coordinating the ubiquitination of spastin. By using vectors encoding non-phosphorylatable or phosphomimetic spastin mutants, S268A and S268D, respectively, we noticed that

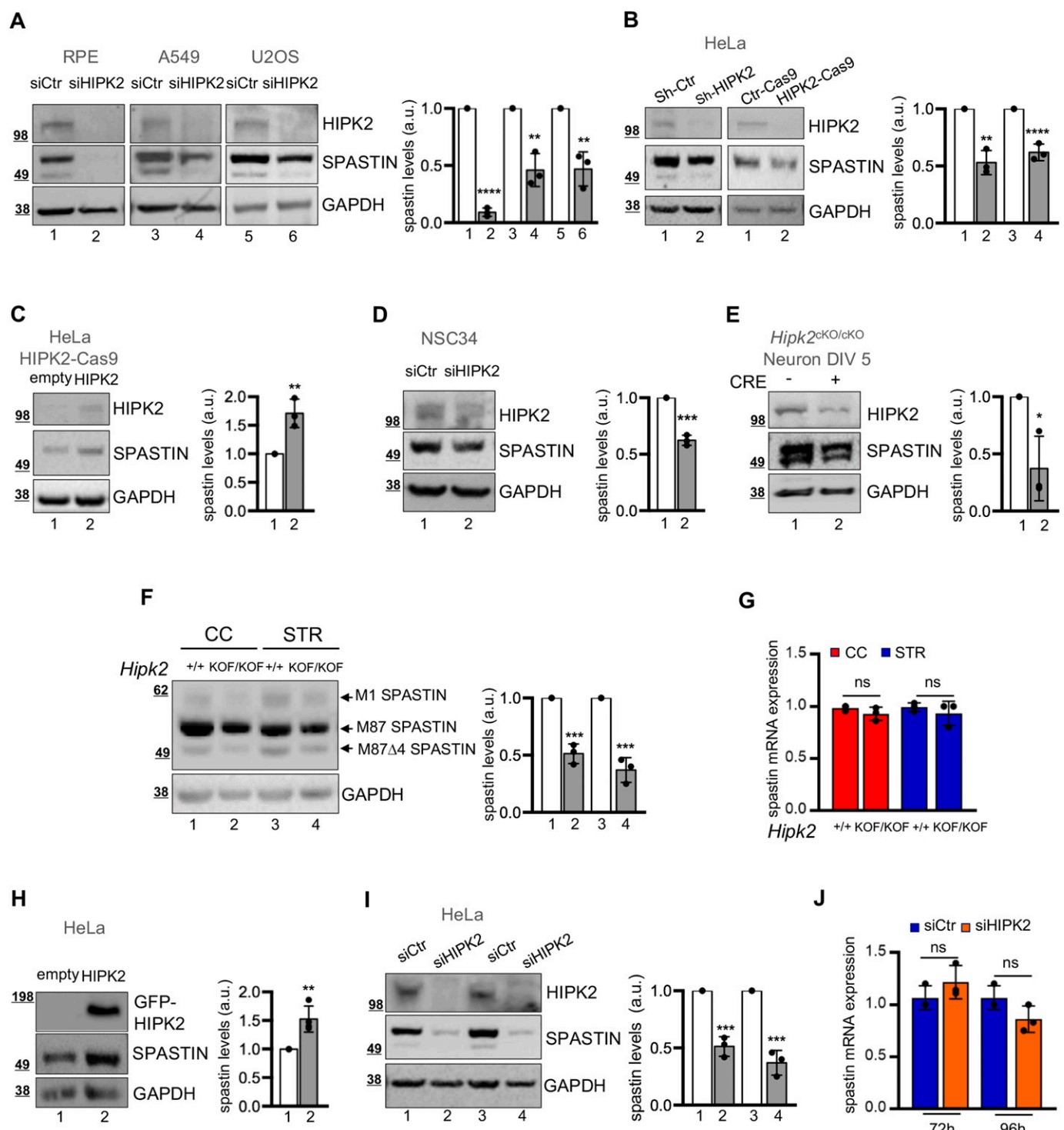

**Figure 1. HIPK2 regulates spastin at posttrascriptional levels.**
**(A)** Indicated cells were transfected with a mix of three human HIPK2-specific (siHIPK2) or negative control (siCtr) stealth siRNAs and analysed by Western blot (WB) 96 h posttransfection. Unless otherwise indicated, here and in the following figures representative WB of three independent experiments was shown and the intensity of the spastin bands quantified, normalized with GAPDH, and reported in a.u. relative to control, as mean ± SD of three independent experiments. Molecular weight markers are reported in kilodalton. Here, spG311/1 Ab was used for spastin detection and statistical differences are relative to corresponding siCtr, unpaired *t* test. **(B)** HeLa cells were transfected with vectors expressing control LacZ sh-RNA (sh-Ctr) or a mix of two HIPK2-specific shRNAs (sh-HIPK2) and analysed by WB 72 h posttransfection (left panels). HeLa HIPK2-Cas9 cells and their parental control cells (Ctr-Cas9) were analysed by WB 24 h after plating (right panels). Statistical differences are relative to corresponding control, unpaired *t* test. **(C)** HeLa HIPK2-Cas9 cells were transfected with low dose of HIPK2-HA–expressing vector to avoid apoptosis induction and analysed by WB 24 h posttransfection with anti-HIPK2 and anti-spastin Abs. Statistical difference is relative to empty vector transfected cells used as control, unpaired *t* test. **(D)** NSC34 cells were transfected with a mix of three mouse HIPK2-specific (siHIPK2) or negative control (siCtr) stealth siRNAs and incubated in differentiation medium;

spastin-S268A was reproducibly and consistently expressed at lower levels than spastin-WT or spastin-S268D when transiently transfected at comparable efficiency (Fig 3B). Accordingly, spastin-S268A displayed a shorter half-life than spastin-WT in the presence of cycloheximide (Fig 3C), indicating that HIPK2-mediated S268 phosphorylation contributes to spastin stability.

### HIPK2-dependent S268 phosphorylation prevents spastin polyubiquitination and degradation

Next, we asked whether S268 phosphorylation is required to prevent spastin polyubiquitination. Thus, we compared on the same blot the polyubiquitination levels among WT, non-phosphorylatable, and phosphomimetic spastin forms in the presence or absence of HIPK2. For clarity, we have presented these data in two separate panels (Fig 4A and B). The non-phosphorylatable spastin-S268A mutant is the most poly-ubiquitinated in HIPK2 proficient Ctr-Cas9 cells (Fig 4A), whereas the phosphomimetic spastin-S268D mutant is the least polyubiquitinated in the HIPK2-Cas9 cells (Fig 4B), in-dicating that HIPK2-mediated S268-phosphorylation prevents spastin polyubiquitination. Accordingly, when transfected at comparable efficiency in HIPK2 proficient and in HIPK2-KO cells, the phosphomimetic spastin-S268D mutant, but not the WT form, is resistant to degradation (Figs 4C and S3). Overall, these findings indicate that HIPK2-mediated S268 phosphorylation protects spastin from polyubiquitination at K554 and degradation.

We next addressed the mechanism by which S268 phos-phorylation regulates ubiquitination. We hypothesised that S268 phosphorylation can protect spastin from polyubiquitination by impairing the recruitment of proteins belonging to the ubiq-uitination pathway or by promoting interactions with factors that mask the domain of spastin necessary for efficient ubiquitination/degradation. To investigate this, we used MS to analyse the interactome of exogenous spastin-S268A and spastin-S268D in HeLa cells (Supplemental Data 1). Among the known interacting proteins, we found atlastin (31), and spastin itself (7, 8, 32). In-terestingly, we found Cullin-Associated NEDD8-Dissociated Pro-tein 1 (CAND1) as a unique spastin-S268D interactor and this preferential interaction was confirmed by co-IP experiments (Fig 4D). Next, we verified spastin/CAND1 endogenous interaction (Fig 4E). According to exogenous co-IP results, we observed that the interaction is stronger in HIPK2 proficient cells than in HIPK2-Cas9 cells, further supporting a preferential interaction between CAND1 and phosphorylated spastin (Fig 4E). CAND1 is a known inhibitor of

Cullin-RING ubiquitin E3-Ligase (CRL) complexes (33), its inter-action with spastin suggests involvement of the neddylation pathway in spastin degradation via UPP.

### Spastin levels and neurite defects can be restored by preventing neddylation-dependent degradation in different *SPG4*-HSP models

To investigate the biological relevance of HIPK2-mediated spastin protein regulation, we analysed the effects of neddylation-mediated UPP inhibition and HIPK2 overexpression on spastin protein levels in pathological conditions due to spastin reduction. We hypoth-esised that as HIPK2 expression, the neddylation inhibitors, such as MLN4924/pevonedistat (33, 34), would stabilize spastin preventing its degradation and subsequently rescue the pathological phe-notypes associated to spastin reduction. To test this hypothesis, first we verified whether MNL4924 treatment stabilizes spastin by preventing its polyubiquitination (Fig S4A and B). Next, we mimicked typical pathological *SPG4*-HSP phenotypes depleting spastin via RNAi in NSC34 differentiated cells. As expected, spastin reduction caused neurite swelling defects (5, 23; Fig S4C) and we observed and quantify these defects (Fig 5B and D, see the light orange columns). In this spastin RNAi neuronal cell model, we observed that HIPK2 overexpression (Fig 5A) or treatment with a low non-toxic dose of MLN4924 (Fig 5C) led to an increase in spastin protein levels and a rescue of neurite swelling (Fig 5A–D), confirming our hypotheses. These results provide evidence that targeting the HIPK2/spastin axis might be a strategy to develop spastin-elevating therapeutic approaches.

Relevant from a clinical point of view, MLN4924 is a CNS-penetrant drug that is currently in several phase 1–3 clinical tri-als for different malignancies (https://clinicaltrials.gov/ct2/results?term=pevonedistat). Thus, we evaluated whether MLN4924 treatment is able to increase the levels of normal spastin allele in the context of pathological heterozygous *SPG4*-HSP mutations. To address this issue, we used two different *SPG4*-HSP models. First, cortical neurons explanted from heterozygous ΔE7 *SPG4* mice and second patient's derived human lymphoblastoid cells carrying heterozy-gous c751insA mutation. Both models express approximately 50% levels of spastin WT and undetectable levels of truncated proteins (23 and data not shown). MLN4924 treatment increased spastin protein levels in both models (Fig 5E and F). Importantly, the in-crease in spastin levels produced by MNL4924 treatment restored levels close to those of control cells (Fig 5E and F), demonstrating that MLN4924 is able to restore para-physiological spastin levels in *SPG4*-HSP haploinsufficient context.

5 d post siRNA transfection cells were analysed by WB. Statistical difference, unpaired *t* test. **(E)** Cortical neurons at 5 days in vitro (DIV5) derived from indicated mouse infected with adenovirus expressing the recombinase CRE were analysed by WB 96 h postinfection. Statistical difference, unpaired *t* test. **(F, G)** Indicated neural tissues were explanted from *Hipk2*[+/+] and *Hipk2*[KOF/KOF] adult mice (5 mo) and spastin levels were analysed by WB and real time RT-PCR. **(F)** Representative WB showing the reduction of spastin isoforms is reported; statistical differences are relative to corresponding control tissue, unpaired *t* test. **(G)** Relative fold-change of spastin mRNA levels, using actin mRNA as normalizer, are represented as mean ± SD of three independent experiments in (G); ns, no statistically significance, unpaired *t* test. **(H)** Representative WB of HeLa cells transfected with GFP (empty) or GFP-HIPK2–expressing vectors and lysed 24 h posttransfection. Statistical difference, unpaired *t* test. **(I, J)** HeLa cells transfected as in Fig 1A were analysed by WB and real-time RT-PCR at the indicated time post transfection. In (I), representative WB, statistical differences are relative to corresponding siCtr, unpaired *t* test; in (J), relative fold-change of spastin mRNA levels, using GAPDH mRNA as normalizer, as mean ± SD of three independent experiments. ns, unpaired *t* test.
Source data are available for this figure.

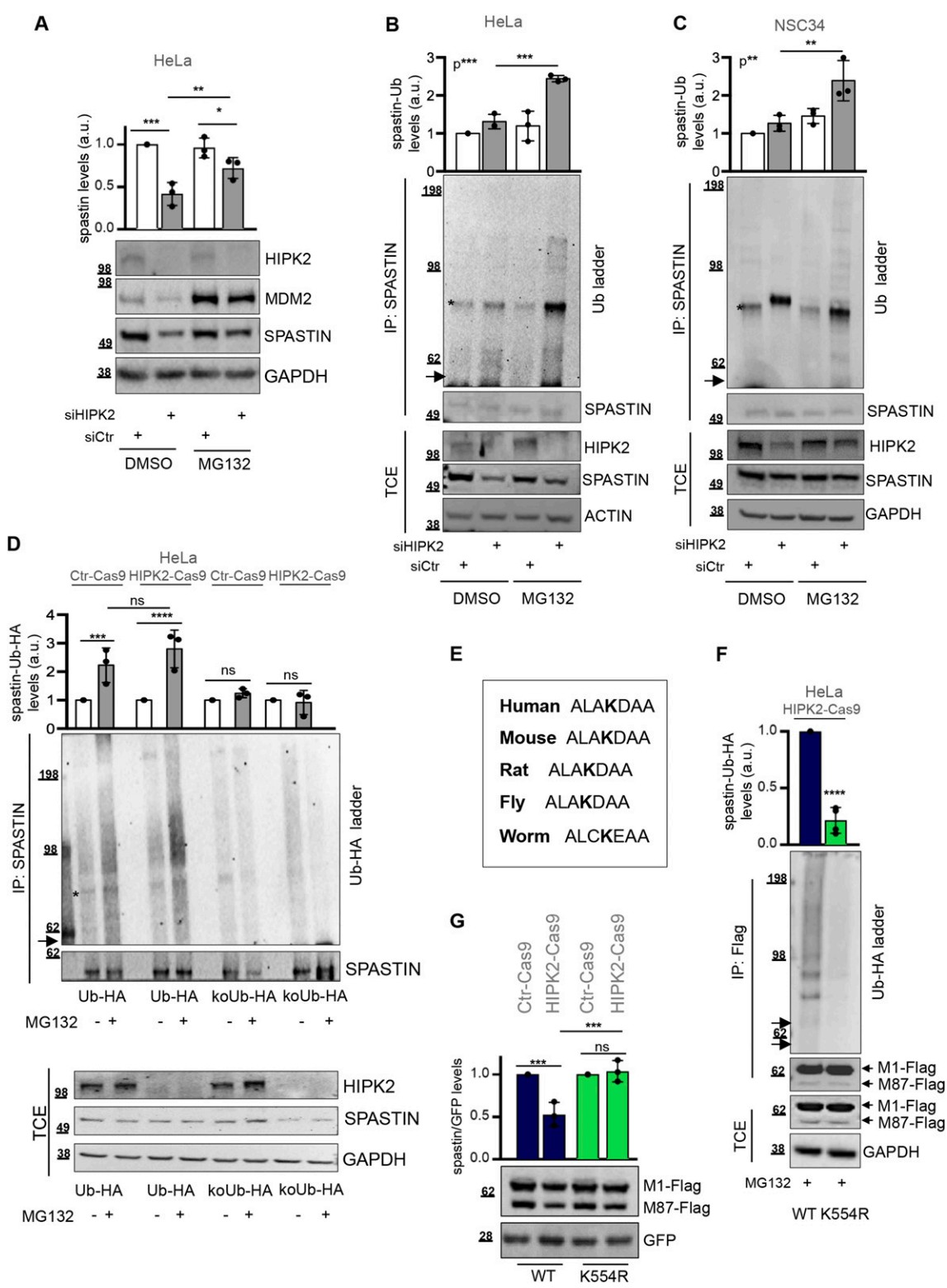

**Figure 2.  HIPK2 regulates spastin via proteasomal degradation through K554 polyubiquitination.**
**(A)** Representative Western blot (WB) of HeLa cells transfected as in Fig 1A and lysed 96 h posttransfection and 8 h after treatment with 20 μM MG132 or its solvent DMSO. MDM2 stabilization has shown as MG132 positive control. Statistical differences, ANOVA test. **(B)** HeLa cells were transfected as in Fig 1A and harvested 96 h posttransfection and 8 h after treatment with 20 μM MG132 or DMSO. Total cell extract (TCE) was analysed by WB for the indicated proteins and immunoprecipitated with anti-spastin Ab. IPs were analysed by WB with anti-Ub and anti-spastin Abs. The arrow indicates the position of the unmodified spastin and the asterisk indicates a nonspecific band. The intensity of spastin-Ub ladder was normalized by the intensity of spastin band in IP and reported relative to siCtr DMSO–treated cells as mean ± SD

# Discussion

We have identified and characterized a pathway responsible for the stability of the spastin protein, which is mutated in the most common form of autosomal dominant HSP, and exploited this knowledge to propose a novel approach for corrective elevation of spastin levels in disease by preventing its neddylation-dependent degradation. In particular, the present study shows that HIPK2-mediated phosphorylation of spastin at S268 inhibits spastin K48-polyubiquitination at K554 and prevents its neddylation-dependent proteasomal degradation, correlating HIPK2-mediated phosphorylation and spastin stability. Our findings support the hypothesis that spastin protein levels are controlled by a balance between the S268 phosphorylation and polyubiquitination/degradation processes (Fig 6).

The precise mechanism by which S268 phosphorylation protects from polyubiquitination remains to be further characterized. The preferential interaction of phosphorylated spastin with CAND1, which functions as substrate receptor exchange factor for CRL1 and 4 complexes (34, 35, 36), supports a model in which the phosphorylation of spastin might alter the dynamic equilibrium with its specific receptor. Therefore, we hypothesized that the enhanced CAND1/spastin interaction might induce a decrease of the exchange possibility necessary for the active CRL complex formation. Nevertheless, we cannot exclude that binders of the p-S268 spastin might act sterically inducing conformational changes that hide site/s for successful ubiquitination. Interestingly, katanin, a MT-severing enzyme belonging to the same ATPase family as spastin, is regulated via phosphorylation/polyubiquitination crosstalk involving similar regions of the protein as we describe for spastin (37, 38, 39), opening the possibility of a more general mechanism for spatio-temporal regulation of ATPases presenting the same domain architecture as spastin and katanin.

Furthermore, because both spastin and HIPK2 have been shown to play roles in different biological processes, our data open the possibility that HIPK2-mediated spastin regulation can function as a fine-tuning regulator in multiple spastin-dependent processes, such as nuclear envelope resealing, cytokinesis and trafficking (1, 3). Recently, spastin severing activity was shown to elicit two different outcomes: the extraction of tubulin subunit from the MT might be counteract by spontaneous incorporation of soluble GTP-tubulin or provoke the severing when extraction outpaces repair (7). We might hypothesize cellular modulation of these two outcomes by locally controlling spastin protein levels acting on the balance between its phosphorylation and ubiquitination/degradation (Fig 6). Further layers of complexity are conceivable considering that spastin levels appear strictly regulated to avoid its accumulation, likely via the action of specific phosphatase/s, deubiquitinase/s, and other factors regulating spastin expression.

Curative therapies and approaches to manage HSP diseases are completely lacking and therapy is only symptomatic. Based on literature evidence (17) and our findings, we hypothesized that elevating spastin by inhibiting its neddylation-dependent degradation might be a novel possible therapeutic approach in SPG4-HSP. The existence of a CNS-penetrant drug, such as MLN4924, that (i) blocks NEDD8-activating enzyme, (ii) is currently in several clinical trials, (iii) has been shown to increase the survival of Spinal Muscolar Atrophy- and Amyothrophic Lateral Sclerosis-derived motorneurons and ameliorate the in vivo phenotype of Spinal Muscolar Atrophy mouse model (40), prompted us to directly assess whether this drug increases the levels of functional spastin in SPG4-HSP haploinsufficient contexts. Our findings indicate that MNL4924 might be a novel spastin-elevating compound, without adverse effects on cell viability.

There are several limitations in this study. We have to consider that neddylation-dependent degradation controls a broad set of proteins and preclinical studies in opportune animal models have to be carried out to assess whether MNL4924 is effective in relieving HSP pathological features. SPG4-HSP shows mild phenotype in homozygote mouse models compared with the severity of the symptoms in HSP patients. The mouse model that we used in this study was an essential mammalian model to be tested for a rescue of spastin levels in neurons. However, its "very weak" swelling phenotype in heterozygous mutant neurons prevents us from assessing the rescue of axon swelling phenotype. Therefore, studies on animal models with a stronger phenotype at the heterozygous status (such as, *Drosophila melanogaster* or *Caenorhabditis elegans*) would be required to fully validate the therapeutic impact of this compound.

Spastin-elevating approaches are not the best therapeutic strategy for the less common missense variants of spastin, acting with pathogenic mechanism other than haploinsufficiency (e.g., dominant-negative or gain-of-function mechanisms). However, it will be interesting to assess the response to UPP of each of these variants. We might speculate to extend this approach to those that

(n = 3). Statistical differences, ANOVA test. **(C)** NSC34 cells were transfected as in Fig 1B and harvested 5 d posttransfection and 8 h after treatment with 20 μM MG132. TCE was analysed by WB for the indicated proteins and immunoprecipitated with anti-spastin Ab. IPs were analysed by WB with anti-Ub and anti-spastin Abs. The arrow indicates the position of the unmodified spastin and the asterisk indicates a specific band. The intensity of spastin-Ub ladder was calculated and reported as in Fig 2B. Statistical differences, ANOVA test. **(D)** HeLa Ctr-Cas9 and HIPK2-Cas9 cells were transfected with vectors expressing HA-tagged Ub-WT (Ub-HA) or its derivative KoUb-HA (i.e., Ub with all lysines mutated in arginines) and treated 24 h posttransfection with 20 μM MG132 or DMSO for 8 h. TCEs were analysed as in Fig 2B. IPs were analysed by WB with anti-HA and anti-spastin Abs. The arrow indicates the position of the unmodified spastin and the asterisk indicates a nonspecific band. The intensity of spastin-Ub-HA ladder was normalized by the intensity of spastin band in IP and reported relative to the correspondent DMSO-treated cells as mean ± SD (n = 3). Statistical differences, ANOVA test. **(E)** Spastin amino acid sequence encompassing the K554 is reported for indicated organisms. Fly = *Drosophila melanogaster*; worm = *Caenorhabditis elegans*. **(F)** HIPK2-Cas9 HeLa cells were transfected with vectors expressing flag-myc–tagged spastin-WT or spastin-K554R in combination with the vector expressing HA-tagged Ub-WT and treated 24 h posttransfection with 20 μM MG132 for 8 h. TCE were analysed by WB and immunoprecipitated with anti-Flag Ab (mouse Ab by Origene Technologies). IPs were analysed by WB with anti-HA and anti-Flag Ab (rabbit Ab by Sigma-Aldrich). The arrows indicate the position of the unmodified spastin isoforms. The intensity of spastin-Ub-HA ladder was normalized by the intensity of spastin bands in IP and reported as mean ± SD (n = 4). Statistical difference, unpaired *t* test. **(G)** HIPK2-Cas9 and Ctr-Cas9 HeLa cells were transfected with vectors expressing spastin-WT or spastin-K554R in combination with peGFP vector at 10:1 molar ratio and analysed by WB 24 h posttransfection. GFP expression was used as internal control for transfection efficiency. Representative WB is shown. The intensity of spastin-Flag bands was normalized by the intensity of GFP and reported relative to correspondent Ctr-Cas9 control cells. Statistical differences, ANOVA test. Source data are available for this figure.

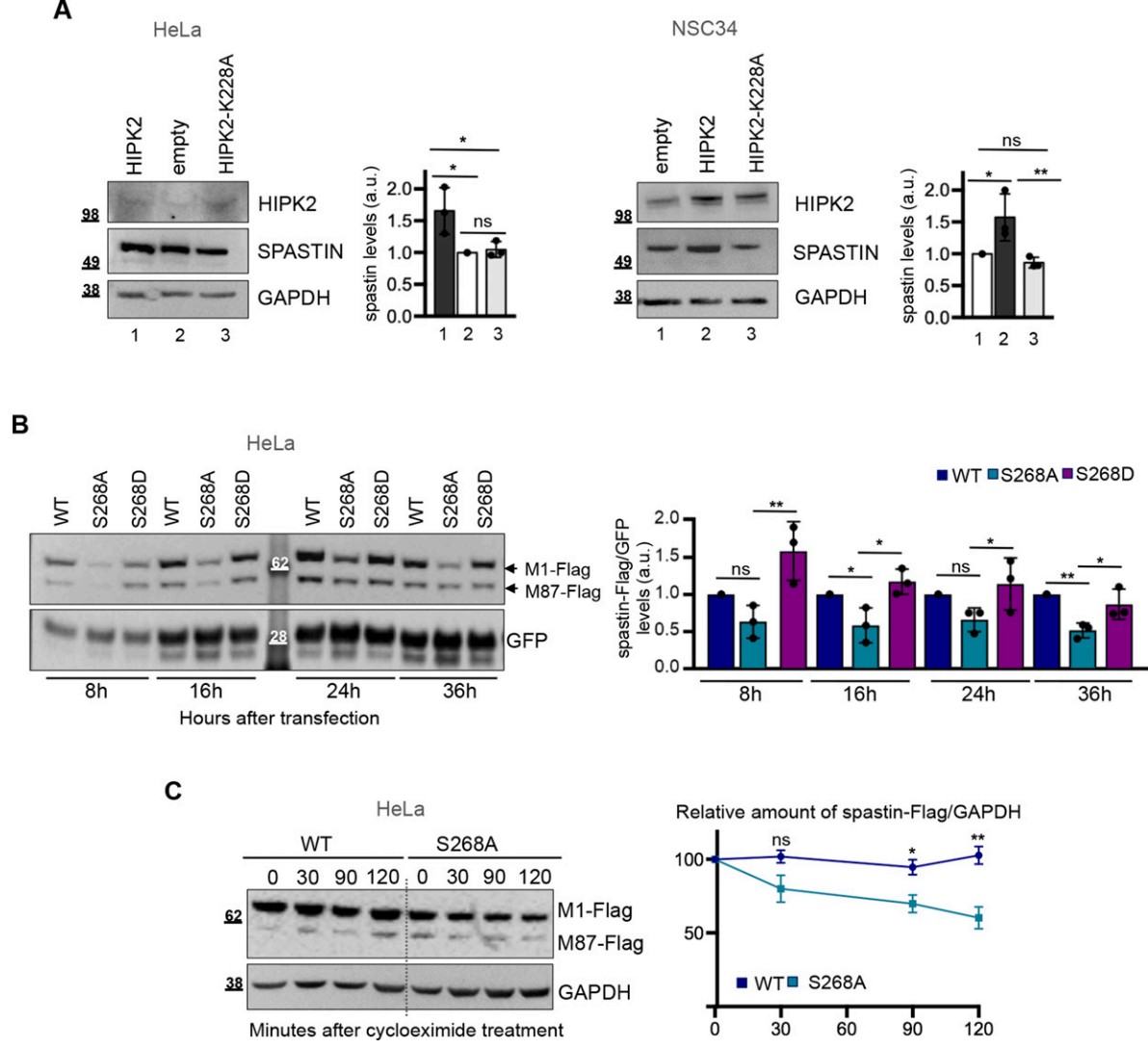

**Figure 3. Kinase activity of HIPK2 regulates spastin protein levels.**
**(A)** Representative Western blot (WB) of HeLa (right panels) and NSC34 cells (left panels) transfected with vectors expressing HA-tagged HIPK2-WT, its derivative -K228A mutant or HA-alone (empty). Statistical differences, Anova test. **(B)** Representative WB of HeLa cells transfected with vectors expressing flag-myc spastin-WT or indicated spastin mutants in combination with peGFP vector at 10:1 molar ratio and lysed at indicated time post transfection. GFP expression was used as internal control for transfection efficiency. The intensity of spastin-Flag bands was normalized by the intensity of GFP and reported relative to spastin-WT for each time point. Statistical differences, ANOVA test. **(C)** HeLa cells were transfected with vectors expressing flag-myc–tagged spastin-S268A or spastin-WT, treated with 25 μg/ml cycloeximide 36 h posttransfection and analysed by WB at indicated times after treatment. Note that to minimize differences in spastin levels at the time 0, cells were transfected with different doses of the expressing vectors, that is, 1 μg of spastin-S268A–expressing vector and 0.5 μg of spastin-WT–expressing vectors. Representative WB is shown. The levels of spastin-Flag bands relative to those of GAPDH were measured at each time point and reported as mean ± SEM of four different independent experiments in the right panel. Statistical differences were calculated and reported for each time point, unpaired t test.
Source data are available for this figure.

show a partial loss of activity triggering degradation. Spastin WT stabilization could alleviate also the pathogenicity of mutants resulting in loss of ATPase activity, such as the I344K variant (41). Preclinical studies with MT-targeting drugs have shown preliminary promising effects for HSP treatment (42, 43), the effects of very low dose of these drugs combined with spastin inhibition of degradation might be also evaluated.

Finally, the possibility to pharmacologically modulate spastin levels opens the way to potential clinical implications also in promoting axon regeneration after nerve injury (6, 44) and in

neurodegenerative diseases associated to spastin dysfunctions, such as Alzheimer's disease (45, 46).

In conclusion, we have identified and characterized a novel pathway regulating spastin protein levels via HIPK2-mediated phosphorylation that prevents spastin neddylation-dependent degradation. Intriguingly, we show that inhibition of neddylation by using the CNS penetrant drug MLN4924 is able to restore spastin levels and rescue neurite defects. Even though many factors have to be considered before initiating preclinical studies for MLN4924 repurposing in *SPG4*-HSP, our novel findings provide proof of

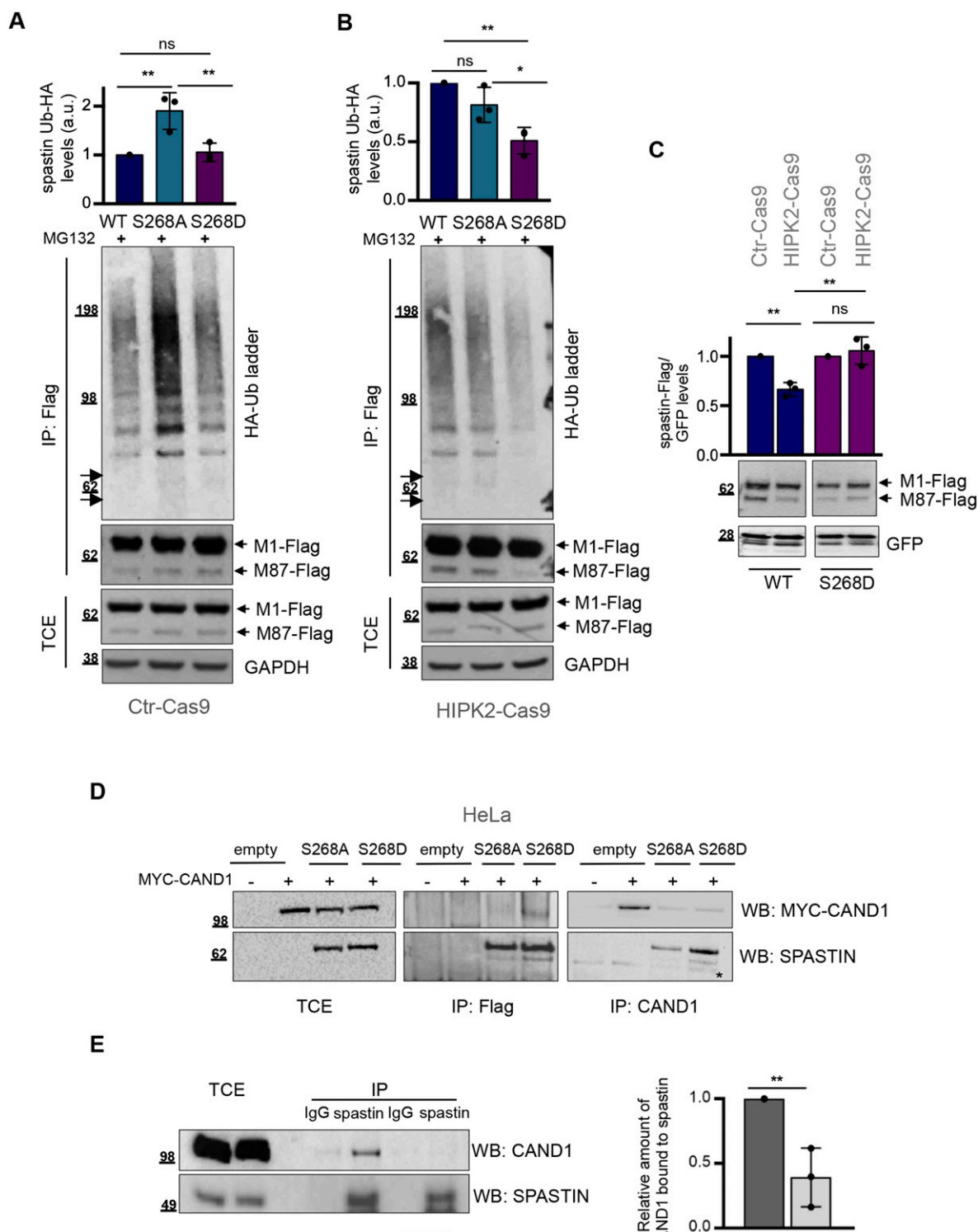

**Figure 4. Phosphorylation in S268 prevents spastin polyubiquitination and degradation.**
**(A, B)** HeLa Ctr-Cas9 (A) and HIPK2-Cas9 (B) cells were transfected with vectors expressing indicated flag-myc–tagged spastin-WT or its derivative mutants in combination with the vector expressing HA-tagged Ub-WT and treated 24 h posttransfection with 20 μM MG132 for 8 h. Total cell extracts (TCEs) were analysed by Western blot (WB) and IP as in Fig 2F. Samples were processed in parallel and analysed on the same blot to make comparison between HIPK2 proficient and null cells. The arrows indicate the position of unmodified spastin isoforms. The intensity of spastin-Ub-HA ladder was normalized by the intensity of spastin-Flag bands in IP and reported as mean ± SD (n = 3). Statistical differences, ANOVA test. **(C)** HeLa Crt-Cas9 and HIPK2-Cas9 cells were transfected with vectors expressing indicated flag-myc–tagged spastin

principle that elevating spastin by inhibiting its UPP-mediated degradation has therapeutic potential for *SPG4*-HSP patients.

# Materials and Methods

### Cells, culture conditions, and treatments

HeLa (a gift of N Corbi), A549 (a gift of R Falcioni), U2OS (a gift of F Moretti), and retinal pigment epithelial (a gift of G Guarguaglini) were cultured at 37°C and 5% $CO_2$ in DMEM GlutaMAX supplemented with 10% heat-inactivated FBS (Life Technologies); mouse moto-neuronal NSC34 (a gift of M Cozzolino) was cultured in DMEM-F12 1:1, supplemented with 10% FBS. Patient-derived lymphoblastoid cells, carrying the pathogenetic heterozygous c751insA *SPG4* mutation, were cultured in RPMI supplemented with 10% FBS. To obtain homogeneous neuronal cell cultures, SNSC34 were differentiated by incubation in serum-free medium supplemented daily with 10 µM retinoic acid (Sigma-Aldrich) and 10 ng/ml GDNF (Peprotech). HeLa HIPK2 null (HIPK2-Cas9) and their parental control (Ctr-Cas9) cells (47) were kindly provided by ML Schmitz. Cells were routinely tested for mycoplasma contamination.

Mice carrying *Hipk2* KO first conditional-ready alleles (*Hipk2*[KOF/KOF]; https://www.mousephenotype.org/data/genes/MGI:1314872) were generated by the International KO mouse consortium by inserting into the *HIPK2* loci a lacZ cassette, which traps and truncates the nascent transcript, leading to 80% of *Hipk2* mRNA reduction. Primary cortical neurons from CRE-inducible *Hipk2* KO alleles (*Hipk2*[cKO/cKO]) were obtained by inbreeding *Hipk2*[KOF/KOF] mice with mice carrying FLP recombinase (https://www.jax.org/strain/005703), and neurons from *SPG4* ΔE7/+ and +/+ mice (23) were prepared and cultured as previously described in reference 23. Adenovirus expressing CRE is a kind gift of D Pajalunga.

The following inhibitors were used: MG132 and cycloeximide (Sigma-Aldrich) and MLN4924 (Cayman Chemical). For the evaluation of cell viability, trypan blue dye exclusion test was used to determine the number of viable cells.

### RNAi and real-time RT-PCR

HIPK2 RNAi was obtained in human cells by using specific stealth siRNAs (a mix of three different validated siRNAs by Life Technologies, as in reference 48) or by transfecting HIPK2-specific sh-RNA–expressing vectors (a mix of pRetroSuper-1376 and -789, (49)). HIPK2 RNAi in murine cells and spastin RNAi in human cells were obtained by using a mix of specific validated stealth siRNAs by Life

Technologies; for spastin, RNAi-specific siRNAs were selected among which targeting sequences common to all spastin isoforms as in reference 26. siRNAs were transfected using Lipofectamine RNAi MAX (Life Technologies). RNA extraction and real-time RT-PCR were performed as in reference 48, and relative fold-change were determined by the $2^{-\Delta\Delta Ct}$ method using GAPDH or actin mRNA as normalizer. All reactions were performed in triplicate, in three independent experiments. Primers for HIPK2 and spastin amplification are as references 48 and 50, respectively. The spastin primers amplify a region common to all spastin isoforms. The sequences of the siRNAs used are the following:

murine-specific siHIPK2 #1: CCACCAACUUGACCAUGACCUUUAA; #2: GAGCCAAGUUCCAACUGGGACAUGA; #3: GCCCAUGUCAAAUCUUGUUUCCAAA. human-specific siHIPK2 #1: CCCGAGUCAGUAUCCAGCCCAAUUU; #2: CCACCAACCUGACCAUGACCUUUAA; #3: CAGGGUUUGCCUGCUGAAUAUUUAU. human/murine-specific siSpastin #1: CCAGUGAGAUGAGAAAUAUUCGAUU, #2 CGGACGUCUAUAACGAGAGUACUAA.

### Expression vectors

The following plasmids were used: peGFP-c2 (Clontech); peGFP-human-HIPK2, flag empty and flag-HIPK2, pET-HA, pET-human-HIPK2-HA, and pET-human-HIPK2-K228A-HA (Kind gifts of ML Schmitz); pSUPER-LacZ and pSUPER-HIPK2-1376 and pSUPER-HIPK2-789 (49), pEF-HA-Ub-WT, and pEF-HA-KoUb (kind gift of A Pollice); pRK5-HA-Ub-K48, pRK5-HA-Ub-WT, and pcDNA3-Myc3-CAND1 (Addgene); Flag-myc empty vector (pCMV6-Entry) and flag-myc–tagged spastin expressing vector (OriGene Technologies). The latter vector expresses higher levels of the M1 isoform than the corresponding M87, in agreement with data showing that M1 is the most abundant form when both isoforms have equally good Kozak's sequences (51). Spastin-S268A and Spastin-S268D mutants were described in reference 26, and spastin-K554R was obtained by site-directed mutagenesis in the spastin-flag-myc–tagged vector using the Quik-Change Lightning Kit (Stratagene) and analysed by sequencing. Vectors were transfected by using Lipofectamine LTX and Plus reagent (Life Technologies).

### Western blot (WB) and immunopecipitation (IP)

Total cell extracts were prepared in RIPA buffer (50 mM Tris–HCl, pH 8, 150 mM NaCl, 0.5% sodium deoxycholate, 0.1% SDS, 1% NP40, and 1 mM EDTA) supplemented with protease and phosphatase inhibitors (Roche). For ubiquitination assays, we first performed WB analysis with anti-spastin or anti-Flag antibodies (Abs) on 1/15 of our IP reactions to determine the efficiency of IP in each sample. After this

WT or its derivative S268D in combination with peGFP vector as in Fig 2G and analysed by WB 24 h post transfection. GFP expression was used as internal control for transfection efficiency. Representative WB is shown. The intensity of spastin-Flag bands was normalized by the intensity of GFP and reported relative to Ctr-Cas9 control cells. Statistical differences, Anova test. **(D)** Representative Co-IP showing that spastin-S268D preferentially binds CAND1. HeLa cells were co-transfected with the plasmid expressing MYC-CAND1 in combination with empty vector or vectors expressing spastin-S268A or spastin-S268D. Cells were collected 24 h posttransfection. TCE were analysed by WB or immunoprecipitated with anti-Flag or anti-CAND1 Abs and analysed as indicated. The asterisk indicates an aspecific band. TCE and IP samples were loaded on the same gel and processed on the same filter. Blots were vertically cropped to show appropriate expositions. Full blots are shown in the source data F4 file. **(E)** Co-IP showing that spastin interaction with CAND1 is stronger in HeLa Ctr-Cas9 cells compared with HIPK2-Cas9 cells. TCE from Ctr-Cas9 and HIPK2-Cas9 cells were analysed by WB or immunoprecipitated with anti-spastin Ab and analysed with indicated Abs. IgG immunoglobulins were used as IP negative control. Band intensities of co-immunoprecipitated CAND1 were normalized by band intensities of spastin immunoprecipitated and their relative values are reported as mean ± SD (n = 3). Statistical difference, unpaired *t* test.
Source data are available for this figure.

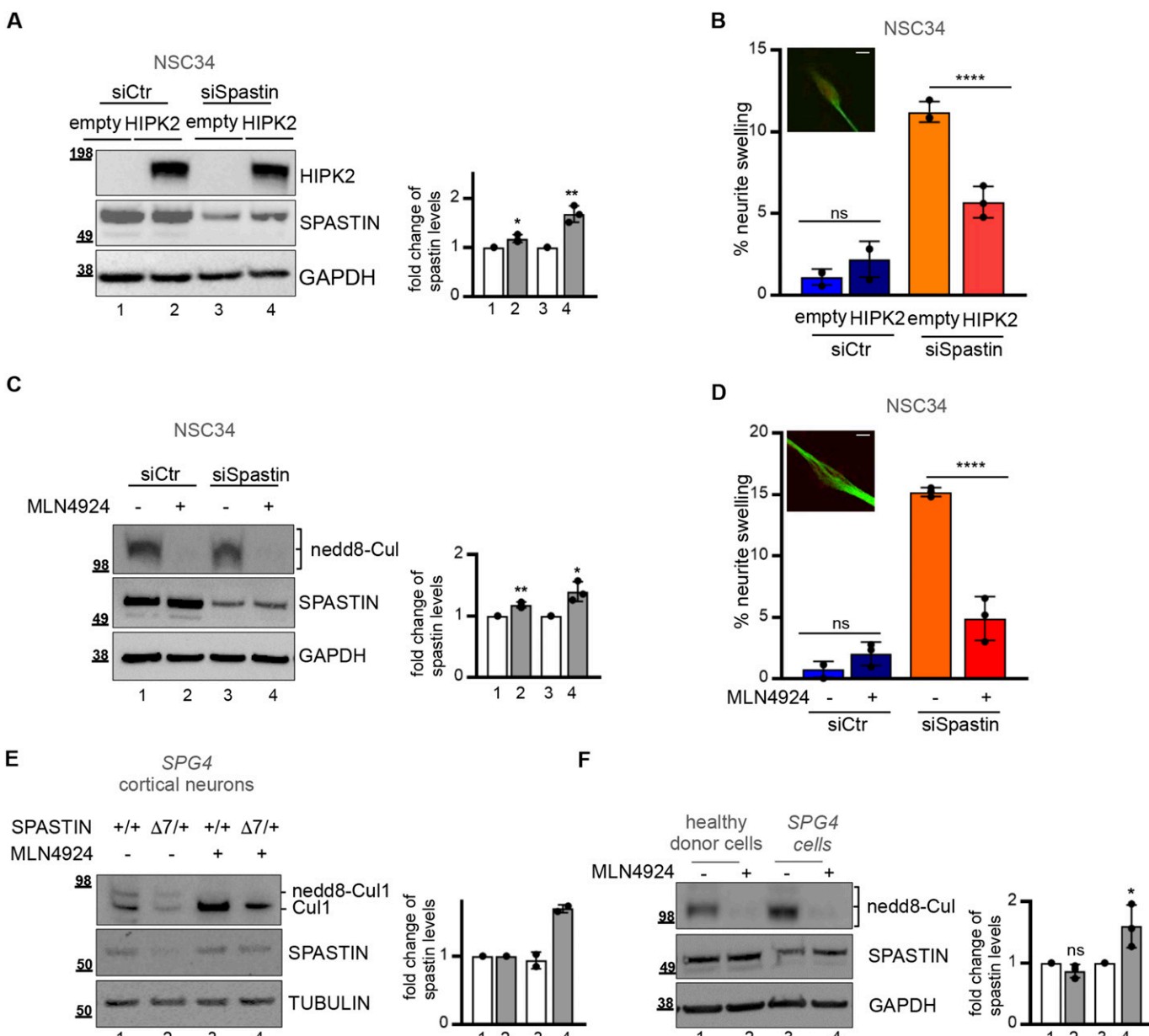

**Figure 5. Restoration of spastin protein levels and neurite swelling in *SPG4*-(HSP) Hereditary Spastic Paraplegia models.**
**(A, B)** NSC34 cells were transfected with 40 nM spastin-specific (siSpastin) or negative control stealth siRNAs and incubated in differentiation medium. 48 h post siRNA transfection cells were transfected with empty or HIPK2 expressing vectors and analysed by Western blot (WB) and by IF using anti-acetylated-tubulin Ab (green) and mitotracker (red) 5 d after siRNA transfection and differentiation. **(A)** Representative WB is shown in (A); the intensity of the spastin bands were quantified, normalized and reported relative to empty vector in siCtr and in siSpastin cells as mean ± SD of three independent experiments. Statistical differences, unpaired *t* test. **(B)** The percentage of swelling is reported in (B) as mean ± SD of three different experiments in which >100 cells were analysed. Statistical differences, ANOVA test. Representative image of neurite swelling is shown. Scale bar, 10 *μ*M. **(A, C, D)** NSC34 cells were transfected as in (A) and incubated in differentiation medium. 72 h post siRNA transfection cells were treated with MLN4924 0.1 *μ*m or its solvent DMSO and analysed 16 h after treatment. **(C)** Representative WB is shown in (C). Cullin deneddylation is shown as MNL4924 positive control. The intensity of the spastin bands were quantified, normalized and reported relative to solvent-treated cells in siCtr and in siSpastin as mean ± SD of three independent experiments. Statistical differences, unpaired *t* test. **(D)** The percentage of swelling is reported in (D) as mean ± SD of three different experiments in which >100 cells were analysed. Statistical differences, Anova test. Representative image of neurite swelling is shown. Scale bar, 10 *μ*M. No dead cells were observed after treatment. **(E)** Representative WB showing the increase of endogenous spastin protein levels 72 h after 0.1 *μ*M MLN4924 treatment in indicated primary cortical neurons derived from *SPG4* WT (+/+) and heterozygous (+/Δ7) E5 mice. The intensity of the spastin bands were quantified, normalized and reported relative to solvent-treated cells in neurons derived from *SPG4* WT (+/+) and heterozygous (+/Δ7) as mean ± SD of two independent experiments. **(F)** Representative WB showing the increase of endogenous spastin protein levels 16 h after 0.1 *μ*M MLN4924 treatment in *SPG4*-HSP lymphoblastoid cells. The intensity of the spastin bands were quantified, normalized and reported relative to solvent-treatment in healthy donors lymphoblastoid cells and in patient's derived *SPG4*-HSP lymphoblastoid cells. Statistical differences, unpaired *t* test.
Source data are available for this figure.

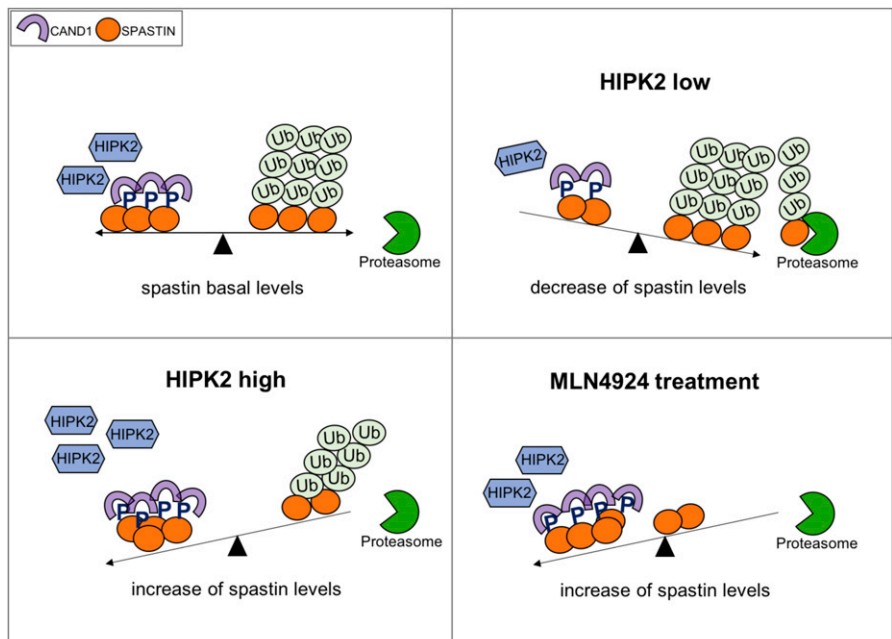

**Figure 6. Schematic model depicting the regulation of spastin protein levels.**
Because S268-phosphorylation (indicated with a blue P) prevents polyubiquitination of spastin, its protein levels are the consequence of a balance between phosphorylation and degradation events depending on HIPK2 kinase activity and the function of a still unknown CRL complex. Based on this model, low HIPK2 activity reduces the spastin phosphorylated forms leading to an increase of spastin polyubiquitination with a consequent decrease of spastin protein levels due to proteasomal degradation. In contrast, high HIPK2 activity protects spastin from polyubiquitination, increasing its protein levels. When polyubiquitination is prevented using the neddylation inhibitor, MLN4924, spastin protein levels increase independently of its phosphorylation status.

quantification, we loaded similar levels of immunoprecipitated spastin and performed WB with anti-ubiquitin (Ub) or anti-HA Abs, depending on the experimental conditions. Proteins were resolved by SDS–PAGE using Bolt Novex Bis-Tris Gels 4–12% gradient or 8% (Life Technologies). Immunoreactivity was determined using ECL-Prime (Amersham), and image acquisition and densitometric analysis were performed with Image Lab software (Bio-Rad). The following antibodies were used: anti-HIPK2 (rat monoclonal Ab C5C6 kindly provided by ML Schmitz); anti-GAPDH #sc-32233 (1:1,000), anti-$\alpha$-tubulin #sc-5286 (1:1,000), anti-vinculin #73614 (1:1,000), anti-spastin mouse monoclonal Abs (1:100; sp3G11/1 #sc-53443 or sp6c6 #sc-81624), and anti-spastin goat polyclonal Ab #sc-49528 (1:1,000) by Santa Cruz Biotechnology (anti-spastin monoclonal Abs were indifferently used because they produce comparable results, whereas anti-spastin polyclonal Ab was used by WB to detect immunoprecipitated spastin in murine cells); anti $\beta$3-tubulin #5568 (1:1,000) and anti-MYC #2276 (1:500) mouse monoclonal Abs (Cell Signaling Technology); anti-CAND1 #NBP1-49918 (1:1,000) rabbit polyclonal Ab (Novus Biological); anti-Ub #BML-PW8810-0100 (1:1,000, clone FK2; Enzo Life Sciences); anti-HA #11583816001 (1:1,000) mouse monoclonal Ab by Roche; anti-GFP (1:800, mouse monoclonal Ab by Roche or 1:200 #sc-390394 rabbit polyclonal Ab by Santa Cruz Biotechnology); anti-Flag (1:1,000; rabbit polyclonal Ab, #F7425, by Sigma-Aldrich or mouse monoclonal Ab, #TA50011; OriGene Technologies); anti-Nedd8 #ab81264 (1:500) rabbit monoclonal Ab by Abcam, anti-Cul1 #612040 (1:500) mouse monoclonal Ab by BD bioscience; and anti–HRP-conjugated goat anti-mouse #7076, anti-rat #7077, and anti-rabbit #7074 (Cell Signaling Technology).

Normal mouse IgG #sc-2025 by Santa Cruz Biotechnology was used as isotype control immunoglobulins in IP experiments.

## MS analysis

24 h after transfection with flag-myc–tagged spastin-S268A or -S268D expressing vectors in HeLa cells extracts were obtained by lysing the cells in RIPA buffer supplemented with protease and phosphatase inhibitors (Roche). IP was performed by incubation of 2 mg of total cell extract with anti-flag Ab (mouse Ab by Origene Technologies) covalent coupled to Dynabeads M-280 Tosylacti-vated according to the according to manufacturer's protocol (Life Technologies). Beads coupled to an unrelated Ab (i.e., anti-E1A mouse monoclonal Ab; BD Bioscience) were used as negative control. Each sample was separated on a 1D-gel NuPAGE 4–12% (Invitrogen), run in MOPS buffer, and stained with the Colloidal Blue Staining kit (Invitrogen). Gel lanes were entirely subdivided into slices and subjected to reduction, alkylation, and trypsin digestion (52). The resulting peptide mixtures were desalted in a trap column (Acclaim PepMap 100 C18, LC Packings; DIONEX), and then separated in a 10-cm reverse-phase column (Silica Tips FS 360-75-8; New Objective) and slurry-packed in house with 5 $\mu$m, 200. Å pore size C18 resin (Michrom BioResources). Separation was performed with an Ultimate 3000 nanoflow HPLC (DIONEX) connected with an LTQ-XL mass spectrometer (ThermoElectron), equipped with a nano-electrospray ion source (ESI).

The list of identified interactors by one experimental replicate has been deposited to the ProteomeXchange Consortium (see the Data Availability section).

## Immunofluorescence (IF) and swelling analysis

Cells were seeded onto poly-L-lysine–coated coverslips, fixed in 2% formaldehyde, permeabilized in 0.25% Triton X-100 in PBS for 10 min, and then blocked in 5% bovine serum albumin in PBS before the primary Ab was applied. The following Abs were used: anti $\beta$3-tubulin #5568 (1:500 mouse monoclonal Abs by Cell Signaling Technology) and anti-acetyl-$\alpha$-tubulin #T7451 (1:500 mouse monoclonal Ab by Sigma-Aldrich). Secondary mouse FITC Ab #A32723 (Alexa-fluor; Life Technologies) was used. DNA was marked with DAPI (Sigma-Aldrich)

and mitochondria with mitotracker Red #M22425 (Life Technologies). Cells were examined under inverted microscope (Eclipse Ti; Nikon) using a Clara camera (ANDOR Technology). Images for each sample were taken in parallel using identical microscope by Nis-Elements H.C. 5.11 using the JOBS module for automated acquisitions. Swelling quantitative analysis was performed as in reference 23. Briefly, swelling was defined along the neurite revealed with acetyl-tubulin and inside mitochondrial staining as larger than 2 µm in diameter; random fields were analysed after staining with mitotracker and the number of swelling scored per at least 100 DAPI stained nuclei.

### Genotyping PCR

Genomic DNA was extracted using DirectPCR lysis reagent (Viagen), and PCR analysis was carried out using GoTaq DNA polymerase (Promega) following the supplier's conditions. Alleles were identified using the following primers: for *Hipk2* wild type (Forward 5′-CGATCGAGTAAGGGTCGGTG-3′, Reverse 5′-GATGTGTGCTTGAGGCTTGC-3′); for *Hipk2 KOF* (Forward 5′-TTATGGTCTGAGCTCGCCATCAGT-3′, Reverse 5′-TGTTTCTCCATGGACAGTGGGTT-3′); for *Hipk2* cKO and KO alleles (distinguishable by different amplicon size; Forward 5′-AAGGCGCATAACGATACCACG-3′, Reverse 5′-CTGTTTCTCCATGGA-CAGTGGGTT-3′).

### Statistical analysis

Data analyses were performed using the GraphPad Prism software. Each experiment has been repeated from three to four times and the results obtained presented as mean ± SD, unless otherwise indicated. The unpaired *t* test or ANOVA test were applied to determine the significance of quantitative experiments. Statistical significance was set at $P < 0.05$ and was reported by asterisks according to the following scheme ****$P < 0.00001$, ***$P < 0.0001$, **$P < 0.001$, and *$P < 0.05$.

### Ethics statement

All mouse experimental procedures were conformed to protocols approved by the Regina Elena National Cancer Institute Animal Care and Use Committee and were performed in accordance with the Guide for the care and use of laboratory animals and the guidelines of the National Institutes of Health, according to the current National Legislation (Art. 31 D.lgs 26/2014, 4 March, 2014) and were approved by National Institutes of Health (1056/2015). The ethics committee of Asl Roma 2, Rome, Italy, approved this study on patient derived cells (0074975/2020). Written informed consent was obtained and data analysis was performed anonymously.

## Data Availability

MS data have been deposited to the ProteomeXchange Consortium via the partner repository PRIDE (53) with the data set identifier PXD021945.

## Supplementary Information

## Acknowledgements

We thank Embo for short-term mobility fellowship to F Sardina. We are grateful to all people cited in the text for their gifts of cells and reagents. We thank Dr F Magi, Dr. B Ciani, and Dr. G Della Zanna for their help and advice. Funding sources: This work was supported by grants from Fondazione Telethon (GGP16072), AFM-Telethon (#22157), Latium Region (85-2017-15348), UK-HSP group, and Italian Association for Cancer Research to C Rinaldo (IG #17739).

### Author Contributions

F Sardina: conceptualization, data curation, formal analysis, funding acquisition, investigation, and writing—review and editing.
A Pisciottani: investigation and methodology.
M Ferrara: investigation.
D Valente: investigation and methodology.
M Casella: investigation and methodology.
M Crescenzi: supervision.
A Peschiaroli: supervision and methodology.
C Casali: resources and methodology.
S Soddu: data curation, supervision, and writing—review and editing.
AJ Grierson: resources, supervision, and writing—review and editing.
C Rinaldo: conceptualization, data curation, formal analysis, supervision, funding acquisition, and writing—original draft.

### Conflict of Interest Statement

The authors declare that they have no conflict of interest.

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
