## [Reviewer comments · Life Science Alliance]

Life Science Alliance

Spastin recovery in hereditary spastic paraplegia by preventing neddylation-dependent degradation

Francesca Sardina, Alessandra Pisciottoni, Manuela Ferrara, Davide Valente, Marialuisa Casella, Marco Crescenzi, Angelo Peschiaroli, Carlo Casali, Silvia Soddu, Andrew Grierson, and Cinzia Rinaldo
DOI: <https://doi.org/10.26508/lsa.202000799>

Corresponding author(s): Cinzia Rinaldo, Institute of Molecular Biology and Pathology (IBPM), National Research Council (CNR) and Francesca Sardina, Institute of Molecular Biology and Pathology (IBPM), National Research Council (CNR)

Review Timeline:

Submission Date:	2020-05-28
Editorial Decision:	2020-06-22
Revision Received:	2020-09-28
Editorial Decision:	2020-10-09
Revision Received:	2020-10-14
Accepted:	2020-10-14

Scientific Editor: Shachi Bhatt

Transaction Report:

June 22, 2020

Re: Life Science Alliance manuscript #LSA-2020-00799-T

Dr. Cinzia Rinaldo
Institute of Molecular Biology and Pathology (IBPM), National Research Council (CNR)
Via degli Apuli,4
Rome, Rome 00185
Italy

Dear Dr. Rinaldo,

Thank you for submitting your manuscript entitled "Spastin recovery in hereditary spastic paraplegia by preventing neddylation-dependent degradation" to Life Science Alliance. The manuscript was assessed by expert reviewers, whose comments are appended to this letter.

As you will see, all referees think that the findings are of interest, but they also have several comments, concerns and suggestions, indicating that a major revision of the manuscript is necessary to allow publication in LSA. As the reports are below, and we think all points need to be addressed, we will not detail them here. Nevertheless, looking at the reports, a major task of the revision should be to improve the quality of the data and to strengthen the study in terms of reproducibility (showing/using more replicates), statistical testing and validation, and quantifications, and by adding missing key controls.

Given the constructive referee comments, we would like to invite you to revise your manuscript with the understanding that all referee concerns must be addressed in the revised manuscript and/or in a detailed point-by-point response. Acceptance of your manuscript will depend on a positive outcome of a second round of review. It is LSAs policy to allow a single round of revision only and acceptance of the manuscript will therefore depend on the completeness of your responses included in the next, final version of the manuscript.

Revised manuscripts should be submitted within three months of a request for revision. We are aware that many laboratories cannot function at full efficiency during the current COVID-19/SARS-CoV-2 pandemic and we have therefore extended our 'scooping protection policy' to cover the period required for full revision. Please contact me to discuss the revision should you need additional time, and also if you see a paper with related content published elsewhere.

Thank you for this interesting contribution to Life Science Alliance. We are looking forward to receiving your revised manuscript.

Sincerely,

Reilly Lorenz
Editorial Office Life Science Alliance
Meyerohofstr. 1
69117 Heidelberg, Germany
t +49 6221 8891 414
e contact@life-science-alliance.org
www.life-science-alliance.org

B. MANUSCRIPT ORGANIZATION AND FORMATTING:

*****IMPORTANT:** It is Life Science Alliance policy that if requested, original data images must be made available. Failure to provide original images upon request will result in unavoidable delays in publication. Please ensure that you have access to all original microscopy and blot data images

before submitting your revision.***

Reviewer #1 (Comments to the Authors (Required)):

In this report the authors characterise a regulatory relationship that they have identified between the microtubule severing protein spastin and the kinase HIPK2. Experiments are presented that suggest that HIPK2 positively regulates spastin expression. The mechanism of this effect is explored, and appears to involve HIPK2 phosphorylation of spastin at residue S268, which it is suggested inhibits K48 poly-ubiquitylation by the ubiquitin ligase CAND1 and subsequent proteasomal degradation of the protein. Furthermore, data is presented to suggest that therapeutic targeting of this pathway could improve neuronal phenotypes in spastin-associated hereditary spastic paraplegia, in which the most common disease mechanism is haplo-insufficiency.

These observations are potentially important for the field, but at present are not robustly supported by the experiments presented, which are in general performed with insufficient rigour - reproducibility of results is not demonstrated (many experiments are presented with n=1), there is minimal statistical treatment of results and key controls are often lacking.

Specific criticisms are as follows:

Figure 1: This figure aims to show that cells or tissues lacking HIPK2 have reduced abundance of spastin. A variety of approaches and cell lines are used, which to a certain extent is a strength, but in many cases the experiments are not performed thoroughly and only n=1 representative blots are shown. To provide confidence in the specificity of the effect, I think it is critical that a rescue experiment should be performed in at least one of the experimental systems (perhaps the HeLa KO cell line would be the most straightforward to use). Throughout this figure the experiments (or at least selected key ones, such as any rescue experiment that is performed, the brain tissue experiments in part H and the HIPK2 induction experiments in J and H) need to be quantified in at least n=3 biological repeats, with the results analysed for statistical significance and presented in a way that allows reproducibility to be assessed (see here for a useful review on this topic: <https://rupress.org/jcb/article/219/6/e202001064/151717/SuperPlots-Communicating-reproducibility-and>). I also don't really understand why the authors used a slightly convoluted approach (adenoviral transduction with Cre-expressing construct) to generate primary neurons partially depleted for HIPK2, when they had a knock-out mouse available - why not just generate knock-out neurons from this? This would have been a cleaner system as the KIPK2 depletion would have been complete (unlike in part G, where residual protein remains). Part H should also show HIPK2 blotting, to validate the knock-out. Part I seems to prefigure the spastin mRNA expression that is also shown in Figure 2 (further comments on this below).

Figure 2: Parts A and B address issues that are also dealt with in Figure 1, and might be better amalgamated into that figure. Interpretation of C) is very difficult, as it is really impossible to know whether the reduced spastin expression in the HIPK2 siRNA expressing cells is an artefact of the experimental system, which combined transient spastin expression with siRNA transfections - in such systems it is very difficult to control for differential expression or cellular toxicity between the different experimental conditions. These experiments would be better conducted in the context of stable cell lines expressing spastin, which would at least remove one variable, or removed from the paper. Part D should show a positive control to validate the effect of MG132. The results in part D also need to be repeated, quantified, analysed statistically and presented as described above for

Figure 1.

Figure 3. Key controls are missing from part B, which should, like part A), show the DMSO only control. Parts C) and D) should show the results for the control wild-type cells, so that the effect of loss of HIPK2 on ubiquitination can also be assessed. Again, the biological reproducibility of the results needs to be demonstrated by quantification, statistical analysis and appropriate graphical presentation of the results.

Figure 4. Again, the biological reproducibility of the results in all three sections (parts A-C) needs to be demonstrated by quantification, statistical analysis for significance and appropriate graphical presentation of the results. This is especially true for part B, as I am not convinced that it is really possible to ensure equal transfection efficiency of the spastin vectors - at least if the same trend was seen in repeat experiments I would have more confidence. In part C), standard deviations for two data points is presented, which really provides no value - n=3 experiments, with appropriate statistical treatment, should be performed.

Figure 5. Part A) - results should be quantified as discussed above. In part D, can the authors show the endogenous co-IP, and whether this is influenced by HIPK2 depletion or over-expression?

Figure 6. Part A - please show the siCtrl/HIPK2 expression result (as shown in the analogous part B). This is a basic control for the experiment. Part B - please show a positive control (i.e. a known target) to validate the effect of the neddylation inhibitor. The effect of the inhibitor on spastin immunoblot dosage does not seem to be reflected in the corresponding quantification (which would suggest a much bigger effect) - please explain. Part C) should show the effect of the drug on control lines.

Minor comments: I would suggest that the Introduction is re-written for the reader who is not expert in this field, who I suspect would find it confusing - for example the explanation of spastin hexamerisation and pore loops is very superficial and the different spastin isoforms are not really explained (e.g. I don't think a reader would really understand what the M1 isoform is from the description provided). The first sentence in the introduction is very convoluted and could be split into two more readable sentences.

I think figures 1 and two could be abbreviated and amalgamated.

Reviewer #2 (Comments to the Authors (Required)):

This is an interesting study that explores the effect of HIPK2 phosphorylation of spastin (already shown by the authors in a previous work) on the protein stability. The authors show decreased levels of spastin in absence of HIPK2, owing to poly-ubiquitination at residue K554, followed by proteasomal degradation. In addition, they find that a spastin phosphomimetic mutant interacts with CAND1, an inhibitor of cullin RING ubiquitin ligases. CAND1 binds unneddylated cullin. The authors therefore propose to use inhibitors of neddylation to increase spastin levels as a potential therapeutic approach. In general, exploring pathways involved in spastin degradation is important, since haploinsufficiency is the pathogenic mechanism in most patients with mutation in SPAST. However, the quality of the data should be improved to fully support the conclusions of the authors.

Specific comments:

1) The data showing the reduction of spastin levels in HIPK2 KO cells (Fig. 1) have been produced in a variety of systems, which shows a general conservation. However, there is variability in the quality of the western blots and it is not clear how many times these experiments have been repeated (the authors refer to representative western blots). The densitometric analysis on one individual blot, which is in some case overexposed is not very informative.

2) The claim that spastin is degraded by the proteasome is based on the one blot in Fig. 2D, which is not convincing. In addition, spastin is not stabilized by MG132 in panel 3A. I think that more experiments with relative quantification are required to fully convince of this result.

3) The pull-down experiments in Figure 3 lack a negative control. The blot showing immunoprecipitated spastin should be shown in full. Would not it be expected to see a signal corresponding to ubiquitinated spastin? The blot in Fig. 3G does not really reflect the quantification.

4) Figure 4: In A, only one spastin band is visible here. Which spastin isoform does this correspond to? In panel B, the authors should show also time point 0, repeat the experiment at least 3 times and quantify. In Fig. 4C, the authors show M1 and M87 in two different panels. This is not convincing. In addition, M87 seems stabilized by MG132 in WT. Overall, the quality of the western blot can be improved.

5) The authors write: "The non-phosphorylatable spastin-S268A mutant is the most polyubiquitinated form in the HIPK2 proficient control cells (Fig. 5A), whereas the phosphomimetic spastin-S268D mutant is the least polyubiquitinated form in the HIPK2-KO cells (Fig. 5B), indicating that HIPK2-mediated S268-phosphorylation prevents spastin polyubiquitination." What can be seen from the two blots is that in both WT and HIPK2 KO cells, the S268D mutant is less polyubiquitinated. The sentence is confusing, and any difference between WT and HIPK2 KO cells is difficult to interpret and quantify since the blots are independent.

6) The authors mention spastin-S268A and S268D pull-down experiments followed by mass spectrometry. It is good scientific practice to show all the data and deposit the results. It is difficult here to evaluate the quality and significance of these data. In addition, the IP showing interaction of spastin and CAND1 are performed following overexpression. It would be interesting to know if CAND1 interacts with endogenous spastin in presence or absence of HIPK2. Furthermore, what is the proposed model exactly? This part remains rather undeveloped.

7) Figure 6: Both in A and B, the claimed rescue of spastin levels following overexpression of HIPK2 or treatment with MNL4924 are not evident. Again, I strongly recommend to repeat this experiment several times and quantify. In addition, the authors should show larger images of the effect of spastin downregulation in NSC34 cells.

Reviewer #3 (Comments to the Authors (Required)):

The study by Sardina et al. investigates the regulation of the stability for the microtubule severing protein spastin. The spastin gene is found mutated in Hereditary Spastic Paraplegia (HSP) that result in reduced spastin levels. The authors, following on their previous observations, they identify a phosphorylation dependent mechanism of spastin stability control. Phosphorylation of spastin by the HIPK2 kinase at serine 268 prevents spastin poly-ubiquitination and subsequent proteasomal degradation. Additionally the authors found that the inhibitor of the ubiquitin-like molecule nedd8

MLN4924 rescues spastin levels and in several functional assays they found that by increasing spastin levels axonal swelling pathology can be reduced.

In general the study is interesting, provides mechanistic insights on spastin stability control with clinical implications. The data are well presented and the great majority are convincing.

While the role of HIPK2-mediated spastin phosphorylation in preventing spastin degradation is well documented, the data linking neddylation and spastin degradation have to be strengthened for the authors to make such conclusions.

As a minimum the authors should directly test the effect of MLN4924 on the spastin half-life and possibly on spastin ubiquitination.

Fig 2A: HIPK2 blot to demonstrate the knockdown

Fig 2B: Which spastin isoforms mRNA levels are presented?

Fig. 3: At least for one experiment a control IP (irrelevant Ab or beads only) lane should be presented.

Fig. 3D: The use of the K48 only mutant is a good experiment but does not exclude that other Ub lysines are involved in the poly-ubiquitination. The use of additional Ub single lysine mutants is recommended.

Fig. 5C: Why is the M87-Flag S268D mutant expressed at much lower levels compared to wild type?

First, we would like to thank the Reviewers for their constructive comments and stimulating questions. We have followed all the Reviewer suggestions and performed all the requested experiments, significantly improving the quality of our Ms. The obtained results have further confirmed our previous data and have strongly enriched the model, that we propose for spastin protein level control. In particular, we presented experiments further verifying the specificity of HIPK2-mediated spastin regulation and characterizing CAND1/spastin interaction in the new Figures 1C and 4E, respectively. We showed new data that clearly links neddylation and spastin degradation in the Figures S3A-B, strengthening the overall message of the Ms. In addition, a model recapitulating our findings on the dynamic balance between phosphorylation and ubiquitination/degradation controlling spastin protein levels has been reported in the new Figure 6.

Here it is a point-by-point response to Reviewer's comments.

Answers to specific questions

Reviewer #1 (Comments to the Authors (Required)):

In this report the authors characterise a regulatory relationship that they have identified between the microtubule severing protein spastin and the kinase HIPK2. Experiments are presented that suggest that HIPK2 positively regulates spastin expression. The mechanism of this effect is explored, and appears to involve HIPK2 phosphorylation of spastin at residue S268, which it is suggested inhibits K48 poly-ubiquitylation by the ubiquitin ligase CAND1 and subsequent proteasomal degradation of the protein. Furthermore, data is presented to suggest that therapeutic targeting of this pathway could improve neuronal phenotypes in spastin-associated hereditary spastic paraplegia, in which the most common disease mechanism is haplo-insufficiency. These observations are potentially important for the field, but at present are not robustly supported by the experiments presented, which are in general performed with insufficient rigour - reproducibility of results is not demonstrated (many experiments are presented with n=1), there is minimal statistical treatment of results and key controls are often lacking.

Specific criticisms are as follows:

Figure 1: This figure aims to show that cells or tissues lacking HIPK2 have reduced abundance of spastin. A variety of approaches and cell lines are used, which to a certain extent is a strength, but in many cases the experiments are not performed thoroughly and only n=1 representative blots are shown. To provide confidence in the specificity of the effect, I think it is critical that a rescue experiment should be performed in at least one of the experimental systems (perhaps the HeLa KO cell line would be the most straightforward to use). Throughout this figure the experiments (or at least selected key ones, such as any rescue experiment that is performed, the brain tissue experiments in part H and the HIPK2 induction experiments in J and H) need to be quantified in at least n=3 biological repeats, with the results analysed for statistical significance and presented in a way that allows reproducibility to be assessed (see here for a useful review on this topic: <https://rupress.org/jcb/article/219/6/e202001064/151717/SuperPlots-Communicating-reproducibility-and>). I also don't really understand why the authors used a slightly convoluted approach (adenoviral transduction with Cre-expressing construct) to generate primary neurons partially depleted for HIPK2, when they had a knock-out mouse available - why not just generate knock-out neurons from this? This would have been a cleaner system as the KIPK2 depletion would have been complete (unlike in part G, where residual protein remains). Part H should also show HIPK2 blotting, to validate the knock-out. Part I seems to prefigure the spastin mRNA expression that is also shown in Figure 2 (further comments on this below).

-In the original MS, we showed one representative WB of at least three different experiments along the entire Ms. We apologize to have not clearly described this point. To make this clearer and more informative, we reported the data quantification and the statistical analysis relative to 3 biological repeats in the Figure 1 and throughout all the experiments reported in the revised Ms, unless otherwise indicated.

-As suggested, to further support the specificity of the HIPK2 depletion effect on spastin levels we show rescue experiment in HeLa KO (HIPK2-Cas9) cells in the new Figure 1C of the revised Ms.

-Reading the reviewer comments about our approach using CRE-expressing construct, we realized that we did not give enough information about our *Hipk2* murine models; we apologize for this. We now clearly describe them in Materials and Methods of the revised Ms. Briefly, we have not *Hipk2* mice totally lacking *Hipk2*, but we have two different *Hipk2* murine models: the CRE-inducible *Hipk2* KO (*Hipk2*^{CKO/CKO}) mice and the *Hipk2* knock-out first allele (*Hipk2*^{KOF/KOF}) mice, expressing a residual 20% *Hipk2* mRNA. We used both to independently support our hypothesis. In particular, in the original Figure 1G (1E in the revised Ms), we used primary neurons derived from *Hipk2*^{CKO/CKO} to generate KO allele after infection with a CRE-expressing adenovirus, whereas in the Figure 1H (1F in the revised Ms), we used brain tissues from *Hipk2*^{KOF/KOF} mice that, beside the residual amount of *Hipk2*, still show a significant reduction of spastin protein levels. For clarity, we now show the HIPK2 residual mRNA levels in the *Hipk2*^{KOF/KOF} brain tissues in the new Figure S1F of the revised Ms. However, we cannot determine the correspondent residual HIPK2 protein expression, because of the low quality of HIPK2 Abs that are able to specifically recognize only high levels of HIPK2 protein in murine tissues.

Figure 2: Parts A and B address issues that are also dealt with in Figure 1, and might be better amalgamated into that figure. Interpretation of C) is very difficult, as it is really impossible to know whether the reduced spastin expression in the HIPK2 siRNA expressing cells is an artefact of the experimental system, which combined transient spastin expression with siRNA transfections - in such systems it is very difficult to control for differential expression or cellular toxicity between the different experimental conditions. These experiments would be better conducted in the context of stable cell lines expressing spastin, which would at least remove one variable, or removed from the paper. Part D should show a positive control to validate the effect of MG132. The results in part D also need to be repeated, quantified, analysed statistically and presented as described above for Figure 1.

- As suggested by the Reviewer, we have amalgamated parts A and B and inserted them in the Figure 1.

- About the experiments reported in Part C, they were originally performed in a controlled manner, to avoid toxicity and verifying transfection efficiency. However, we agree with the Reviewer, it is not easy to monitor all the variables. Unfortunately, stable spastin overexpression cannot be induced, because it leads to cytoskeleton MT severing, cell detachment, and death. As also acknowledged by the Reviewer, we originally showed comparable results by expressing exogenous spastin in HeLa-Ctr-Cas9 and HIPK2 KO (HIPK2-Cas9) cells (see WB and blue columns in Figures 2G and 4C of the revised Ms), thus, we removed the original Figure 2C from the paper.

-As suggested by the Reviewer about Part D (2A in the revised Ms), we have added a positive control to validate MG132 effect (*i.e.*, MDM2 immunodecoration) and data are now presented with quantification and statistical analysis relative to 3 biological repeats.

Figure 3. Key controls are missing from part B, which should, like part A), show the DMSO only control. Parts C) and D should show the results for the control wild-type cells, so that the effect of loss of HIPK2 on ubiquitination can also be assessed. Again, the biological reproducibility of the results needs to be demonstrated by quantification, statistical analysis and appropriate graphical presentation of the results.

As requested, parts B, C and D (2C, 2D and S2C, respectively, in the revised Ms) show the controls (i.e., DMSO and control wild-type cells) and are presented with data quantification and statistical analysis relative to 3 biological repeats.

Figure 4. Again, the biological reproducibility of the results in all three sections (parts A-C) needs to be demonstrated by quantification, statistical analysis for significance and appropriate graphical presentation of the results. This is especially true for part B, as I am not convinced that it is really possible to ensure equal transfection efficiency of the spastin vectors - at least if the same trend was seen in repeat experiments I would have more confidence. In part C), standard deviations for two data points is presented, which really provides no value - n=3 experiments, with appropriate statistical treatment, should be performed.

As requested, we presented Fig. 4A, B (3A, B in the revised Ms) with data quantification and statistical analysis relative to 3 biological repeats. Part 4C (3C in the revised Ms) has been presented with data quantification relative to 4 biological repeats.

Figure 5. Part A) - results should be quantified as discussed above. In part D, can the authors show the endogenous co-IP, and whether this is influenced by HIPK2 depletion or over-expression?

In the revised Ms, we present Fig. 5A, B (now 4A, B) with data quantification and statistical analysis relative to 3 biological repeats. In the new Figure 4E, we now show the endogenous co-IP. In addition, prompted by the Reviewer suggestions, we observed that CAND1-spastin interaction is stronger in HIPK2 proficient cells (Ctr-Cas9) than in HIPK2 null cells (HIPK2-Cas9), further supporting a preferential interaction between CAND1 and phosphorylated spastin. These findings were discussed and reported also in the final model in the new Figure 6.

Figure 6. Part A - please show the siCtrl/HIPK2 expression result (as shown in the analogous part B). This is a basic control for the experiment. Part B - please show a positive control (i.e. a known target) to validate the effect of the neddylation inhibitor. The effect of the inhibitor on spastin immunoblot dosage does not seem to be reflected in the corresponding quantification (which would suggest a much bigger effect) - please explain. Part C) should show the effect of the drug on control lines.

Part A - as suggested, we now show the requested controls (i. e, the siCtrl/HIPK2 expression data and the positive control for MNL4924 activity). In particular, we show reduction of cullin neddylated forms after MNL4924.

Reading the Reviewer's comments about the effect of inhibitor in part B, we realized that we did not clearly describe the lanes that we compared: we apologize for this. We have now clarified this issue in the figure legend and make it clear in the graph. In particular, the MNL4924 effect on spastin was independently reported relative to DMSO-treated cells in siCtr (lanes 1-2) and in siSpastin cells (lanes 3-4). Data quantification and the statistical analysis relative to 3 biological repeats is now reported and clearly described in the revised Ms.

As suggested, in part C, we now show the effect of the drug on control cells established from healthy donors.

- Note that figure 6 is figure 5 in the revised Ms.

Minor comments: I would suggest that the Introduction is re-written for the reader who is not expert in this field, who I suspect would find it confusing - for example the explanation of spastin hexamerisation and pore loops is very superficial and the different spastin isoforms are not really explained (e.g. I don't think a reader would really understand what the M1 isoform is from the description provided). The first sentence in the introduction is very convoluted and could be split into two more readable sentences. I think figures 1 and two could be abbreviated and amalgamated.

We have followed these suggestions.

Reviewer #2 (Comments to the Authors (Required)):

This is an interesting study that explores the effect of HIPK2 phosphorylation of spastin (already shown by the authors in a previous work) on the protein stability. The authors show decreased levels of spastin in absence of HIPK2, owing to poly-ubiquitination at residue K554, followed by proteasomal degradation. In addition, they find that a spastin phosphomimetic mutant interacts with CAND1, an inhibitor of cullin RING ubiquitin ligases. CAND1 binds unneddylated cullin. The authors therefore propose to use inhibitors of neddylation to increase spastin levels as a potential therapeutic approach. In general, exploring pathways involved in spastin degradation is important, since haploinsufficiency is the pathogenic mechanism in most patients with mutation in SPAST. However, the quality of the data should be improved to fully support the conclusions of the authors.

Specific comments:

1-The data showing the reduction of spastin levels in HIPK2 KO cells (Fig. 1) have been produced in a variety of systems, which shows a general conservation. However, there is variability in the quality of the western blots and it is not clear how many times these experiments have been repeated (the authors refer to representative western blots). The densitometric analysis on one individual blot, which is in some case overexposed is not very informative.

We originally showed one representative WB of at least 3 different experiments along the entire Ms. We apologize to have not described clearly this point. To make this clearer and more informative, we reported the data quantification and the statistical analysis relative to 3 biological repeats throughout all the experiments in the revised Ms, unless otherwise indicated. Overexposed individual blots were substituted with lower exposition. To further support the specificity of the HIPK2 depletion effect on spastin levels we show also rescue experiment in HeLa KO (HIPK2-Cas9) cells in the new Figure 1C of the revised Ms.

2-The claim that spastin is degraded by the proteasome is based on the one blot in Fig. 2D, which is not convincing. In addition, spastin is not stabilized by MG132 in panel 3A. I think that more experiments with relative quantification are required to fully convince of this result.

To make spastin degradation by the proteasome more convincing, we reported the data quantification and the statistical analysis relative to 3 biological repeats in the figure 2A of the revised Ms. In the panel 3A, that is panel 2B in the revised Ms, we show improved quality spastin and actin blots relative to TCE. Data quantification of the relative spastin levels shows spastin stabilization after MG132- in these TCEs, as reported in the Supportive Figure 1, presenting data quantification relative to 3 biological repeats. We did not include this TCE quantification data in the revised Ms. However, if the Reviewer and the Editors consider it appropriate, we can show them.

3-The pull-down experiments in Figure 3 lack a negative control. The blot showing immunoprecipitated spastin should be shown in full. Would not it be expected to see a signal corresponding to ubiquitinated spastin? The blot in Fig. 3G does not really reflect the quantification.

As suggested, we show negative IP controls in the new Figures S3 A,B of the revised Ms. The blots showing immunoprecipitated spastin do not present clear signal corresponding to ubiquitinated spastin when immunodecorated with anti-spastin Ab. This is not uncommon when using Abs against the protein of interest, because the epitope might be blocked or masked by the presence of ubiquitin chains, preventing the Ab from recognising the modified version of the protein (Emmerich and Cohen, 2015).

The Figure 3 G (Fig 2G in the revised Ms) reports the quantification data of spastin-Flag levels relative to GFP (our internal efficiency transfection control) on 3 different biological replicates. To make the data more evident, we report quantification by using bar graphs with individual data points in addition to error bars and other statistical information (see below for considerations on spastin-Flag level quantification).

4) Figure 4: In A, only one spastin band is visible here. Which spastin isoform does this correspond to? In panel B, the authors should show also time point 0, repeat the experiment at least 3 times and quantify. In Fig. 4C, the authors show M1 and M87 in two different panels. This is not convincing. In addition, M87 seems stabilized by MG132 in WT. Overall, the quality of the western blot can be improved.

-In the original Fig 4A, the spastin band is M87, that is the most abundant of spastin isoforms. The M87 delta exon 4 is not always easy to detect because it is less abundant and has only a slightly different size compared with M87. By following the Reviewer's suggestion, in the revised Ms, we show a higher-quality representative WB out of the 3 independent experiments performed- where both spastin isoforms are visible and presented data quantification and statistical analysis relative to 3 biological replicates.

-In B, the time is "hours after transfection"; at the zero time point, there is no expression of exogenous proteins. To make this clearer, we included the axis title below the blot and, as requested, presented data quantification and statistical analysis relative to 3 biological replicates.

-By reading reviewer comments about spastin isoforms in C, we realized that it is crucial to show a unique blot with both M1 and M87 to appreciate their relative levels. Indeed, we have to point out that our spastin expression vectors express both M1 and M87 isoforms, but the majority of spastin was expressed as M1. This is in agreement with data showing that when start codons of both isoforms had equally good Kozak's sequences allowing equally efficient translation, M1 always accumulated significantly more than corresponding M87 (Solowska et al., 2014). This information has been reported in the Materials and Methods section of the revised Ms. Based on the above considerations, we reported spastin-Flag levels relative to M1+M87 for each time point in B, in C, and along the entire revised Ms, whenever exogenous spastin was used. In addition, we showed improved quality blots and presented data quantification relative to 4 independent experiments in the revised Ms. We apologize for the lower quality of the previous blot that led the reviewer to consider M87-spastin WT stabilized by Chx.

Note that figure 4 is now figure 3 in the revised Ms.

5) The authors write: "The non-phosphorylatable spastin-S268A mutant is the most polyubiquitinated form in the HIPK2 proficient control cells (Fig. 5A), whereas the phosphomimetic spastin-S268D

mutant is the least polyubiquitinated form in the HIPK2-KO cells (Fig. 5B), indicating that HIPK2-mediated S268-phosphorylation prevents spastin polyubiquitination." What can be seen from the two blots is that in both WT and HIPK2 KO cells, the S268D mutant is less polyubiquitinated. The sentence is confusing, and any difference between WT and HIPK2 KO cells is difficult to interpret and quantify since the blots are independent.

We apologize for the misunderstanding. Indeed, the blots were not independent but belong to the same Western blot. However, we realized that we did not include this information in the text. We have now clarified this issue. In addition, the blot is reported in full in the source files.

6) The authors mention spastin-S268A and S268D pull-down experiments followed by mass spectrometry. It is good scientific practice to show all the data and deposit the results. It is difficult here to evaluate the quality and significance of these data. In addition, the IP showing interaction of spastin and CAND1 are performed following overexpression. It would be interesting to know if CAND1 interacts with endogenous spastin in presence or absence of HIPK2. Furthermore, what is the proposed model exactly? This part remains rather undeveloped.

As requested, we show all the data in the supplementary file S1. Furthermore, stimulated by the Reviewer's suggestions, we verified spastin/CAND1 endogenous interaction. In addition, we observed that spastin/CAND1 interaction is stronger in HIPK2 proficient cells than in HIPK2 null cells, further supporting a preferential interaction between CAND1 and phosphorylated spastin. These findings have been shown in the new Figure 4E of the revised Ms. CAND1 functions as substrate receptor exchange factor for CRL1 and 4 complexes (Liu et al., Mol cell 2018; Reichermeier et al., Mol cell 2020). The preferential interaction of phosphorylated spastin with CAND1 supports a model in which the phosphorylation of spastin might alter the dynamic equilibrium with its specific receptor. The latter is a key step for the efficient formation of an active CRL complex. Therefore, we hypothesized that the enhanced spastin/CAND1 interaction might induce a decrease of the exchange possibility necessary for the active CRL complex formation. This model has been proposed in the Discussion, without excluding the possibility that binders of the p-S268 spastin might act also sterically inducing conformational changes that hide site/s for successful ubiquitination. Nevertheless, the precise mechanism by which S268 phosphorylation protects from polyubiquitination remains to be further characterised.

7) Figure 6: Both in A and B, the claimed rescue of spastin levels following overexpression of HIPK2 or treatment with MNL4924 are not evident. Again, I strongly recommend to repeat this experiment several times and quantify. In addition, the authors should show larger images of the effect of spastin downregulation in NSC34 cells.

As suggested, data quantification and statistical analysis relative to 3 biological repeats is reported in the revised Ms. In the new Figure S3C, we show large representative fields of NSC34 control and spastin-depleted cells. Arrows were used to indicate neurite swelling. Note that the Figures 6A and B are Figures 5A and B in the revised Ms.

Reviewer #3 (Comments to the Authors (Required)):

The study by Sardina et al. investigates the regulation of the stability for the microtubule severing protein spastin. The spastin gene is found mutated in Hereditary Spastic Paraplegia (HSP) that result in reduced spastin levels. The authors, following on their previous observations, they identify a phosphorylation dependent mechanism of spastin stability control. Phosphorylation of spastin by the

HIPK2 kinase at serine 268 prevents spastin poly-ubiquitination and subsequent proteasomal degradation. Additionally, the authors found that the inhibitor of the ubiquitin-like molecule nedd8 MLN4924 rescues spastin levels and in several functional assays they found that by increasing spastin levels axonal swelling pathology can be reduced. In general, the study is interesting, provides mechanistic insights on spastin stability control with clinical implications. The data are well presented and the great majority are convincing.

While the role of HIPK2-mediated spastin phosphorylation in preventing spastin degradation is well documented, the data linking neddylation and spastin degradation have to be strengthened for the authors to make such conclusions.

As a minimum the authors should directly test the effect of MLN4924 on the spastin half-life and possibly on spastin ubiquitination.

As suggested by the Reviewer, we tested the MNL4924 effect on spastin half-life showing that it is higher in MNL4924 treated cells compared to solvent-treated cells. In addition, we showed that MNL4924 treatment strongly inhibits spastin polyubiquitination, further supporting data linking neddylation to spastin degradation. These new data were reported in the Figures S3A, B of the revised Ms.

Fig 2A: HIPK2 blot to demonstrate the knockdown.

We showed HIPK2 blot in this Figure, that is 1J in the revised Ms.

Fig 2B: Which spastin isoforms mRNA levels are presented?

As reported in the Materials and Methods section, the spastin primers amplify a region common to all spastin isoforms. To make this clearer, we indicated it in the figure legend.

Fig. 3: At least for one experiment a control IP (irrelevant Ab or beads only) lane should be presented.

We showed control IPs with IgG for endogenous and exogenous spastin experiments in the new Figures S2A, B.

Fig. 3D: The use of the K48 only mutant is a good experiment but does not exclude that other Ub lysines are involved in the poly-ubiquitination. The use of additional Ub single lysine mutants is recommended.

We have addressed this issue by using, in parallel to K48-only mutant, the K63-only mutant that drives a signal commonly linked to “proteasome-independent” processes. As shown in the Supportive Figures 2A,B, spastin-Ub-HA signals are unambiguously detectable in Ub-HA and K48-Ub-HA expressing cells after MG132 treatment. Instead, it is difficult to detect a consistent and reproducible spastin-Ub-HA signal in K63-Ub HA expressing cells and, as expected, no significant changes were observed after MG132 treatment in these cells. Even if we cannot exclude the involvement of other non-proteolytic processes in spastin regulation, we moved forward to investigate the proteasomal degradation pathway. We did not include K63-Ub data in the original manuscript. However, if the Reviewer and the Editors consider it appropriate, we can show these data and the relative Figure in the revised Ms.

Fig. 5C: Why is the M87-Flag S268D mutant expressed at much lower levels compared to wild type?

In this figure, we aim to compare the levels of S268D mutant in HIPK2 proficient (Ctr-Cas9) and in HIPK2-KO (HIPK2-Cas9) cells and those of spastin-WT in HIPK2 proficient and in HIPK2-KO (HIPK2-Cas9) cells. However, as shown in the Supportive Figure 2C, if we compared the levels of spastin WT and S268D, we do not observe statistically significant difference in HIPK2 proficient cells (Ctr-Cas9). We did not include this type of comparison in the original Ms. However, if the Reviewer and the Editors consider it appropriate, we can show this comparison in the revised Ms.

Supportive Figure 1

Data quantification of spastin/loading control levels by WB on TCE, relative to data shown in Figure 2B of the revised MS. Quantification was performed as in 2 A. Bars are mean \pm SD of three independent experiments; $p=0.0085$, Anova test

Supportive Figure 2

(A,B) No consistent and reproducible spastin-Ub-HA signals are detectable in K63-Ub-HA transfected cells.

Indicated cells were transfected with K63-Ub-HA expressing vectors in parallel with K48-Ub-HA and Ub-HA ones. 24h post transfection cells were treated with 20 μ M MG132 or DMSO for 8h. TCE were analysed by WB and IP with indicated Abs. In **A**, representative WB of three biological repeats is shown. The arrow indicates the position of the unmodified spastin and the asterisk indicates a non-specific band. In **B**, data quantification reporting spastin-Ub-HA levels relative to spastin immunoprecipitated was presented as mean \pm SD of three independent experiments.

K63-Ub-HA data are highlighted with a yellow box. The K48-Ub-HA and Ub-HA experiments (lanes 1-4) are shown in the Figure S2C,D of the revised Ms.

Comparison of spastin-Flag WT and S268D levels in HeLa Ctr-Cas9 cells, data quantification relative to Figure 4C. Spastin-Flag intensity relative to our internal efficiency transfection control (GFP) was reported relative to M87-spastin-Flag, M1-spastin-Flag and to spastin-Flag (i.e., M1+ M87).

October 9, 2020

RE: Life Science Alliance Manuscript #LSA-2020-00799-TR

Dr. Cinzia Rinaldo
Institute of Molecular Biology and Pathology (IBPM), National Research Council (CNR)
Via degli Apuli,4
Rome, Rome 00185
Italy

Dear Dr. Rinaldo,

Thank you for submitting your revised manuscript entitled "Spastin recovery in hereditary spastic paraplegia by preventing neddylation-dependent degradation". We would be happy to publish your paper in Life Science Alliance pending final revisions necessary to meet our formatting guidelines, and the statistics and text changes requested by the reviewers (comments at the end of this email).

Along with the requests below, please also address the following in your revised manuscript:

- please add ORCID ID for both corresponding authors-you should have received instructions on how to do so
- please add a callout for Figure 1D&F, Figure 4D, Figure 5C in your main manuscript text
- please make sure that the order of the manuscript sections are in accordance with LSA's guidelines (<https://www.life-science-alliance.org/manuscript-prep#format>)
- please deposit the Mass Spec data in a public database and provide the accession number in the manuscript under 'Data Availability' section (<https://www.life-science-alliance.org/manuscript-prep#datadepot>)
- please include the Supporting figures 1 and 2C, included in the pbp rebuttal, in the revised manuscript
- no need to add supportive figure 2B in the revised manuscript, but instead we encourage you to point out in the revised manuscript that involvement of other non-proteolytic processes in spastin regulation cannot be over-ruled.

A. FINAL FILES:

B. MANUSCRIPT ORGANIZATION AND FORMATTING:

Sincerely,

Shachi Bhatt, Ph.D.

Executive Editor
Life Science Alliance
<https://www.life-science-alliance.org/>
Tweet @SciBhatt @LSAJournal

Reviewer #1 (Comments to the Authors (Required)):

The authors are to be congratulated for producing a vastly improved version of this manuscript, in which they have diligently addressed all of my main points. I think this paper now tells an interesting and convincing story, which will be of significant interest to the field. I would make a few minor points:

1. When ANOVA is used, the significance of differences in key individual columns should also be compared. This would be relevant to figures 2b, 2c and 3b. This is important as, for example in Figure 2B and C, we are really interested in the difference between columns 2 and 4. This is very straightforward using 1-way anova in graphpad prism, which I note is the software the authors are using.
2. Figure 2D - are the differences between columns 2 and 4 significant? It looks like they will not be and the authors should word the description of this carefully, so as not to make any unsupported claims.
3. Figure 2G. Columns 2 and 4 should be compared statistically.
4. Figure 4A and 4B. As a general comment, where more than two columns are compared in one experiment, 1-way anova with analysis of individual pairwise comparison should be used, instead of multiple t-tests.
5. In figure 5 the interpretation of the immunoblot quantification histograms is a little complex, as they refer to normalised intensities that are then reported relative to one of two controls. The Y-axis label doesn't quite reflect this, and at first glance it would be easy to mistake what is being shown as simple quantification of the blots (I only spotted this because there is such a mismatch between the blot intensities and the histogram column sizes). Perhaps the y-axis label could be altered to better reflect the reality of what is being shown. I'm not actually convinced that this way of representing these data is the clearest, so perhaps the authors could consider a more conventional way of representing them.

Two very minor points:

1. In the introduction, it is claimed that M1 spastin is only present in the neuronal cells. Although this claim has appeared in the literature before, it is simply not true. M1 spastin has been demonstrated in many different cell types, albeit at low abundance.
2. Page 2, line 60, I think the authors should also mention the possibility of a dominant negative effect for spastin missense mutants. In my opinion this is the most likely pathological mechanism for these.

Reviewer #2 (Comments to the Authors (Required)):

The authors have substantially revised the manuscript, providing biological replicates and quantification of the main experiments and adding crucial controls.

The text should be carefully checked, since in several places the reference to the Figures and the actual figures did not fit. Examples: Fig. 1 panels are wrongly cited. Figure 4 D, E in the text are in fact Fig. 5D and E.

MS experiments were performed only once, and it is not clear if a negative control for the IP was used. Several identified proteins are in fact likely contaminants. A note of caution on this experiment shall be noted.

Reviewer #3 (Comments to the Authors (Required)):

n/a

We have followed all the Editor and Reviewer suggestions. Here it is a point-by-point response to the comments.

Along with the requests below, please also address the following in your revised manuscript:

-please add ORCID ID for both corresponding authors- **DONE**

-please add a callout for Figure 1D&F, Figure 4D, Figure 5C in your main manuscript text **DONE**

-please make sure that the order of the manuscript sections are in accordance with LSA's guidelines **DONE**

-please deposit the Mass Spec data in a public database and provide the accession number in the manuscript under 'Data Availability' section.

DONE The account details for Reviewer are:

Username: reviewer_pxd021945@ebi.ac.uk

Password: bfKGvh2s

-please include the Supporting figures 1 and 2C, included in the pbp rebuttal, in the revised manuscript **DONE**

-no need to add supportive figure 2B in the revised manuscript, but instead we encourage you to point out in the revised manuscript that involvement of other non-proteolytic processes in spastin regulation cannot be over-ruled. **As suggested, we pointed it out at page 4 lane 136 of the revised Ms**

Reviewer #1 (Comments to the Authors (Required)):

The authors are to be congratulated for producing a vastly improved version of this manuscript, in which they have diligently addressed all of my main points. I think this paper now tells an interesting and convincing story, which will be of significant interest to the field. I would make a few minor points:

1. When ANOVA is used, the significance of differences in key individual columns should also be compared. This would be relevant to figures 2b, 2c and 3b. This is important as, for example in Figure 2B and C, we are really interested in the difference between columns 2 and 4. This is very straightforward using 1-way anova in graphpad prism, which I note is the software the authors are using. **DONE**
2. Figure 2D - are the differences between columns 2 and 4 significant? It looks like they will not be and the authors should word the description of this carefully, so as not to make any unsupported claims. **The Reviewer is right, the differences are not statistically significant. In the revised Ms we have already carefully described this point (see page 4 lanes 130-131). Now, we have pointed it also in the Figure.**
3. Figure 2G. Columns 2 and 4 should be compared statistically. **DONE**

4. Figure 4A and 4B. As a general comment, where more than two columns are compared in one experiment, 1-way anova with analysis of individual pairwise comparison should be used, instead of multiple t-tests. **DONE**
5. In figure 5 the interpretation of the immunoblot quantification histograms is a little complex, as they refer to normalised intensities that are then reported relative to one of two controls. The Y-axis label doesn't quite reflect this, and at first glance it would be easy to mistake what is being shown as simple quantification of the blots (I only spotted this because there is such a mismatch between the blot intensities and the histogram column sizes). Perhaps the y-axis label could be altered to better reflect the reality of what is being shown. I'm not actually convinced that this way of representing these data is the clearest, so perhaps the authors could consider a more conventional way of representing them. **We have changed the y-axis label for easier reading.**
6. Two very minor points: 1. In the introduction, it is claimed that M1 spastin is only present in the neuronal cells. Although this claim has appeared in the literature before, it is simply not true. M1 spastin has been demonstrated in many different cell types, albeit at low abundance. **We have removed this claim.** 2. Page 2, line 60, I think the authors should also mention the possibility of a dominant negative effect for spastin missense mutants. In my opinion this is the most likely pathological mechanism for these. **As suggested, we mentioned also this possibility.**

Reviewer #2 (Comments to the Authors (Required)):

The authors have substantially revised the manuscript, providing biological replicates and quantification of the main experiments and adding crucial controls.

The text should be carefully checked, since in several places the reference to the Figures and the actual figures did not fit. Examples: Fig. 1 panels are wrongly cited. Figure 4 D, E in the text are in fact Fig. 5D and E. **We apologize for this. We have checked and corrected them.**

MS experiments were performed only once, and it is not clear if a negative control for the IP was used. Several identified proteins are in fact likely contaminants. A note of caution on this experiment shall be noted. **In the original Ms we have described in the Material and methods section that we used an unrelated Ab as negative control. Now, we added a note about the experimental replicate in the same section (see page 13 lane 423-424).**

Reviewer #3 (Comments to the Authors (Required)):n/a

October 14, 2020

RE: Life Science Alliance Manuscript #LSA-2020-00799-TRR

Dr. Cinzia Rinaldo
Institute of Molecular Biology and Pathology (IBPM), National Research Council (CNR)
Via degli Apuli,4
Rome, Rome 00185
Italy

Dear Dr. Rinaldo,

Thank you for submitting your Research Article entitled "Spastin recovery in hereditary spastic paraplegia by preventing neddylation-dependent degradation". It is a pleasure to let you know that your manuscript is now accepted for publication in Life Science Alliance. Congratulations on this interesting work.

DISTRIBUTION OF MATERIALS:

Again, congratulations on a very nice paper. I hope you found the review process to be constructive and are pleased with how the manuscript was handled editorially. We look forward to future exciting submissions from your lab.

Sincerely,

Shachi Bhatt, Ph.D.

Executive Editor

Life Science Alliance

<https://www.life-science-alliance.org/>
